# The effect of cyclones crossing the Mediterranean region on sea level anomalies at the Mediterranean Sea coast

Piero Lionello[1,2], Dario Conte[2], and Marco Reale[3,4]

[1]Università del Salento, Di.S.Te.B.A., via per Monteroni 164, Lecce
[2]CMCC , Centro Euro-Mediterraneo per i Cambiamenti Climatici
[3]ICTP, Ex Sissa Building, Trieste, Italy
[4]OGS, Sgonico, Trieste, Italy
**Correspondence:** Piero Lionello (piero.lionello@unisalento.it)

**Abstract.** Large positive and negative sea level anomalies at the coast of the Mediterranean Sea are linked to intensity and position of cyclones moving along the Mediterranean storm track, with dynamics involving different factors. This analysis is based on a model hindcast and considers nine coastal stations, which are representative of sea level anomalies with different magnitude and characteristics. When a shallow water fetch is present, the wind around the cyclone center is the main cause of sea level positive and negative anomalies, depending on its onshore or offshore direction. The inverse barometer effect produces a positive anomaly at the coast near the cyclone pressure minimum and a negative anomaly at the opposite side of the Mediterranean Sea. The latter is caused by the cross-basin mean sea level pressure gradient that is associated to the presence of a cyclone. This often coincides with the presence of an anticyclone above the station, which causes local negative inverse barometer effect. Further, at some stations, negative sea level anomalies are reinforced by a residual water mass redistribution within the basin, which is associated with a transient response to the atmospheric pressure forcing. Though the link with the presence of a cyclone in the Mediterranean has comparable importance for positive and negative anomalies, the relation between cyclone position and intensity is stronger for the magnitude of positive events. Area of cyclogenesis, track of the central minimum and position at the time of the event differ depending on the location where the sea level anomaly occurs and on its sign. The western Mediterranean is the main cyclogenesis area for both positive and negative anomalies, overall. Atlantic cyclones mainly produce positive sea level anomalies in the western basin. At the easternmost stations, positive anomalies are caused by cyclogenesis in the Eastern Mediterranean. North Africa cyclogenesis is a major source of positive anomalies at the central African coast and negative anomalies at the eastern Mediterranean and North Aegean coast.

# 1 Introduction

On the synoptic time scale characterizing the evolution of weather conditions, the combined action of wind and mean sea level pressure (MSLP) causes temporary deviations of sea level from its mean value. These sea level anomalies (SLAs) typically grow and decay on a temporal scale ranging from few hours to days.

In the Mediterranean region (MR) the complex morphology, the land-sea distribution and the large heat and moisture fluxes provided by the Mediterranean Sea, lead to a peculiar branch of the mid-latitude storm track, which has important effects on the environment (Lionello et al. (2006a)). The MR is, indeed, characterized by high frequency of cyclogenesis processes, in particular in areas such as the lee of the Alps and of the Atlas mountains, the North Aegean Sea, Cyprus area, the Black Sea,

and the Iberian Peninsula. The presence of the Mediterranean branch of the North Hemisphere storm track is evident in the results of cyclone tracking methods (Lionello et al (2016); Flaounas et al. (2016)) and it has been extensively studied in the scientific literature (H.M.S.O. (1962); Alpert et al. (1990); Trigo et al. (1999); Maheras et al. (2001); Trigo et al. (2002); Lionello et al. (2002); Nissen et al. (2010); Garcies and Homar (2011); Campins et al. (2011) among others).

Several studies have investigated the link between cyclones and severe weather events in the MR, such as extreme precipitation, windstorm, floods, landslides (Lionello et al. (2006a); Nissen et al. (2010); Lionello et al. (2012); Ulbrich et al. (2012); Pinto et al. (2013); Reale and Lionello (2013)). However, studies that have described the evolution of the synoptic weather conditions leading to storm surges in the MR have focused on single specific locations or areas, e.g. Ullmann et al. (2007) for the Camargue coast. Many studies (Robinson et al. (1973); Trigo et al. (2002); Lionello (2005); Lionello et al.

(2012); Međugorac et al. (2015, 2018)) have shown that floods in the Northern Adriatic coast are associated with cyclones moving along the North Atlantic storm track or secondary cyclones triggered by these systems in the north-western MR. Other studies have shown that at the Mediterranean coast of France, storm surges are induced by Atlantic storms entering the MR following a northwest/southeast direction (Moron and Ullmann (2005)). Except papers describing meteo-tsunamis (e.g. Vilibić and Šepić (2009); Monserrat et al. (2006); Šepić et al. (2015)) there is little literature considering the synoptic conditions

leading to storm surges at other locations and no study has considered negative SLAs. However, several studies have modelled the evolution of storm surges as driven by MSLP (mean sea level pressure) and surface wind fields (Marcos et al. (2011); Conte and Lionello (2013); Androulidakis et al. (2015); Makris et al. (2016); Vousdoukas et al. (2016); Lionello et al. (2017)).

This study investigates the link of both positive and negative large SLAs along the Mediterranean coastline to the passage

of cyclones over the region (figure 1) and describes how SLAs evolve and respond to the presence of cyclones. It includes an analysis of the dynamics of SLAs, of the synoptic patterns associated with them and of the variations of these patterns with the position where SLA occurs. It aims at contributing arguments for understanding the link between the variability and evolution of the MR storm track and of SLAs. It describes position and track of cyclones that are associated with extreme SLA and shows

the link between their intensity and the magnitude of the corresponding SLAs.

Section 2 describes the data and methodology used. Section 3, first, analyses the reliability of the model hindcast to represent the synoptic conditions leading to large SLAs. Successively, it investigates characteristics and tracks of cyclones leading to large SLAs in the MR, and the link between cyclones (specifically their position and intensity) and SLAs. Further the dynamics of the modelled response of SLAs to the passage of cyclones is described. The final section 4 contains a short discussion and describes the main conclusions of this study.

## 2 Data and Methods

Nine coastal stations (figure 1, Alicante, Gabes, Toulon, Tripoli, Trieste, Dubrovnik, Iskenderun, Alexandria, Thessaloniki) have been selected on the basis of a former study (Conte and Lionello (2013)), which has described the distribution of the largest SLAs along the coast of the Mediterranean Sea and the relative importance of the wind and sea level pressure forcing. These nine stations are meant to be representative of SLAs with different magnitude and characteristics. Trieste is representative of the North Adriatic Sea and Gabes of the homonymous gulf, which are the two areas in the MR where the action of the wind produces the largest SLAs. Toulon, Thessaloniki and Tripoli are representative of large SLAs, along the Mediterranean coast of France, the northern Aegean Sea and along the southern Mediterranean coast, where wind and sea level pressure have comparable importance. Iskenderun, Alexandria and Alicante are stations where SLAs are smaller than in the other considered stations and the importance of the wind action is small. At Dubrovnik, along the eastern Adriatic coast, SLAs have large values and the relative importance of sea level pressure is larger than in the northern Adriatic.

This analysis is based on a hindcast, which has been carried out with a barotropic ocean circulation model based on depth-averaged currents (Lionello (2005)). The HYPSE model domain covers the whole Mediterranean Sea with an open boundary (located west of the Gibraltar Strait) that connects the Mediterranean Sea to the Atlantic Ocean, along which a constant (zero) sea level is imposed. The model lat-lon regular grid adopts a $0.05^O$ resolution in this hindcast and it has been recently developed into multi-core version, named HYPSE (Hydrostatic Parallel Surface Elevation). This sea level hindcast (1979-2001, hereafter COSMO-ERA) has been forced using the MSLP and surface wind fields provided at a $0.12^O$ resolution by a downscaling of the ERA-Interim reanalysis (Dee et al. (2011)) that has been carried out with the COSMO-CLM (Rockel et al. (2008)) model. The hindcast has already been validated in Conte and Lionello (2013). The simulation describes well the large SLA values in Northern Adriatic sea and describe the difference between this sub-regional peak and the rest of the coastline. Unfortunately, lack of data prevent model validation along the African coast. In general the model underestimates large SLAs, with a tendency to perform worse in the western Mediterranean than along the rest of the coast and to perform percent-wise better for negative than positive SLAs. In this study we could use hourly tide gauge data for validation only in four of the stations considered in this study (Alicante, Toulon, Trieste, Dubrovnik). Percent rms error on large SLAs is less than 10% for Toulon and Trieste, and

in the range 30-40% for Alicante and Dubrovnik.

Time series of observed hourly sea level values at four tide gauges (Alicante, Toulon, Trieste, Dubrovnik) have been used to identify observed extremes and to discuss the reliability of the hindcast to reproduce the synoptic conditions leading to large

5 SLAs. These four series are a subset of those used in Conte and Lionello (2013) (and, previously, in Jordà et al. (2012) and Marcos et al. (2009)) for the validation of the HYPSE model. Here, they have been selected because of their location and length, which, except for Alicante, whose data stops in 1998, covers the whole COSMO-ERA hindcast and offers the possibility of discussing the phenomenology of large SLAs in the western Mediterranean. In the eastern Mediterranean Sea and along the coast of Africa, time series sufficiently long and regular for computing adequate statistics are not available to the authors.

The procedure to compute the SLAs consists of two steps:

a) both observations and model results have been preprocessed using a HPF (High-Pass Filter) with a cutoff frequency of 1/30 days$^{-1}$ (Conte and Lionello (2013)) in order to cancel long term components (due to change of mass of the Mediterranean Sea and steric effects) and to isolate the component that is caused by the short term meteorological forcing.

b) the intensity of individual events has been computed extracting the overall maximum (minimum) over a 120 hour-long time window. Successive events are considered independent if the separation between the maxima (minima) is longer than 120 hours[1]. The 100 largest positive and 100 largest negative SLAs in the time series have been extracted from both simulated and observed time series and used for the analysis.

The selection of 100 hundred events is a subjective decision. Considering that the hindcast covers 22 years, this corresponds

to an approximate average of 5 events per year. In the case of Venice this is close to the 80th percentile of the surge events (Lionello et al. (2012)). Empirical tests have shown that results do not appreciably change using a smaller sample. Before applying these two steps to the observed time series, the astronomical tide component had been subtracted by means of a harmonic analysis (the standard program t-tide, Pawlowicz et al. (2002) has been used). The COSMO-ERA hindcast does not include the astronomical tide.

Cyclones are identified using the cyclone tracking method that is described in Lionello et al. (2002) and Reale and Lionello (2013). This method, which has been compared to other tracking procedures in the IMILAST project (Intercomparison of Mid-Latitude Storm Diagnostics, Neu et al. 2013,Ulbrich et al. (2013); Lionello et al (2016)), is based on the search of pressure minima in MSLP gridded fields. It identifies the location where each cyclogenesis process occurs and constructs the

30 trajectory of the pressure minimum by joining the location of the low-pressure center in successive maps until it disappears (cyclolysis). This cyclone tracking algorithm contains features that are meant to detect the formation of cyclones inside the Mediterranean and, at the same time, to avoid the inflation of the number of cyclones, determined by considering small, short

---

[1]This period has been selected to ensure independence of the events. considering the whole Mediterranean region, Lionello et al (2016) show that cyclones lasting more than 5 days are extremely rare. Considering the specific situation of the Adriatic Sea, it is meant to avoid the superposition with seiches triggered by previous events, which have a period of about 22 hours and an attenuation of about 10% at each cycle (Lionello et al. (2006b).

lived features as independent systems. This is a crucial balance as a large fraction of Mediterranean cyclones are secondary lows triggered by the presence of a large system over north and central Europe. The method first partitions the SLP field in depressions, which can be considered candidates for independent cyclones, by merging all steepest descent paths leading to the same minimum. The small depressions that share a boundary with a deeper depression are included in the latter to form a single

cyclone. The position of the cyclone is computed as the average of the points with SLP not more than 3 hPa higher than the actual minimum to compensate for large deviation of cyclones from the circular shape. Finally, when searching for successive positions of cyclones to construct their track, the search area is shifted southeasterly with respect to the former center (see Lionello et al. (2002) and Reale and Lionello (2013) for more details). Following Neu et al. (2013), cyclone tracking has been performed using the MSLP fields of the global ERA-Interim analysis, which consists of 6-hourly fields at 0, 6, 12 and 18 UTC,

covering the period 1979-2001. The result is a list of cyclones with the position of their center in geographical coordinates and the corresponding MSLP minimum and depth[2] (both in hPa). It was not possible to apply the tracking to the COSMO-ERA MSLP, because the limited domain of this hindcast does not allow to identify cyclogeneses occurring outside the MR.

## 3  Results

### 3.1  Agreement between mean sea level pressure composites of observed and simulated events

Initially we discuss whether the synoptic patterns that are associated with large SLAs in the COSMO-ERA hindcast are similar to those occurring when large SLAs are observed. Necessarily, this comparison is limited to the 4 stations where observed time series are available.

In general, the ranking of SLAs in the observed and simulated time series differs substantially. Consequently, the list of the

100 largest observed ("OBS") and of the 100 largest simulated ("MOD") events share only a fraction of events (table 1). The small number of common events is explained by the grouping of the largest SLAs in a relatively narrow range of values, so that small differences in their magnitude may correspond to large differences in their rank. Therefore, inaccuracies of the HYPSE model and of the driving meteorological fields imply substantial differences in ranking between observed and simulated SLAs.

Figure 2 shows MSLP composites corresponding to large positive SLAs. Each map results from the average of the 6-hourly ERA-Interim MSLP fields that are closest to the time of the 100 largest peak of independent SLAs, considering the "OBS" (left column) and "MOD" lists (right column). In both columns the MSLP composites show the presence of cyclones. Trieste and Dubrovnik exhibit similar synoptic patterns, have a large number of events common to the "MOD" and "OBS" lists (50% and 49% in the table 1 ) and the highest value of spatial correlation (0.98) between the corresponding MSLP composites. Slightly

lower values of correlation (0.89 / 0.91) and a smaller number of common events (29% / 23%) occur at Alicante and Toulon. These correlation values are computed between the "MOD" and "OBS" MSLP composite including all grid points within a

---

[2]The depth of a cyclone is an estimate of the differences between the pressure minimum and the surrounding background value (see Reale and Lionello (2013) for details on its computation)

| Location | Positive anomalies | | Negative anomalies | |
| --- | --- | --- | --- | --- |
| | percent of common events | spatial correlation between "OBS" and "MOD" | percent of common events | spatial correlation between "OBS" and "MOD" |
| Alicante | 29% | 0.89 | 25% | 0.96 |
| Toulon | 23% | 0.91 | 21% | 0.96 |
| Trieste | 50% | 0.98 | 32% | 0.98 |
| Dubrovnik | 49% | 0.98 | 27% | 0.90 |

**Table 1.** Observed and modelled sea level anomalies: for each location the table reports the fraction of events that are common to the lists of observed and simulation events, and the spatial correlation between the corresponding SLP composites.

distance of 20 grid steps from the station. Autocorrelation within a MSLP field is high and the estimate of the actual degrees of freedom would be required for assessing the significance of these correlation values. Performed tests show that significance remains above the 95% confidence level even if a number of degree of freedom as low as 20 is assumed.

Figure 3 shows MSLP composites corresponding to large observed and modelled negative SLAs. Also for negative SLAs values of spatial correlation between "MOD" and "OBS" composites are large (never lower than 0.90). The number of events common to the two lists is lower for negative than for positive SLA, correspondingly suggesting larger errors in the hindcast than for positive SLAs. Note that the composites of Alicante and Trieste report the presence of a cyclone over the Mediterranean Sea also in association with negative SLAs.

In general, cyclones are deeper and more well defined in the "MOD" composites than in the "OBS" composites. Actually, if SLAs are extracted from the model hindcast, the model simulation ensures a strong consistency between the MSLP fields (contributing to the "MOD" composites) and SLAs. If SLAs are extracted from observed time series, the corresponding MSLP (contributing to the "OBS" composite") might incorrectly reproduce real position and the intensity of the cyclones, because of errors in the ERA-Interim reanalysis and in the COSMO-ERA hindcast.

The similarity between the two columns of figure 2 and 3 show that the MSLP pattern associated to large positive SLAs in the model simulation is accurate and realistic, in spite of inaccuracies in the simulated maxima of individual events. Consequently, we conclude that the MSLP composites based on simulated events reproduce realistically the atmospheric patterns leading to large SLAs.

### 3.2 Evolution of the MSLP field during the development of large sea level anomalies

The panels of figures 4 and 5, based on the 100 largest positive and negative SLA, respectively, show the ERA-Interim MSLP composites 48 hours (left column), 24 hours (mid column) before and at the time (right column) of the SLA maxima. Each

line of figures 4 and 5 considers separately one of the nine coastal stations. Figure 4, which considers positive SLAs, shows the presence of a cyclone, which is consequently a permanent feature in the atmospheric circulation leading to large positive SLAs. Also figure 5, which considers negative SLAs, for most stations shows the presence of a cyclone in the basin, except for Dubrovnik, Thessaloniki and Tripoli.

Locations along the northern coastline (Toulon, Trieste, Dubrovnik, Thessaloniki) for positive SLAs and, to a lesser extent, along the southern coastline for negative SLAs (Iskenderun, Alexandria, Gabes) evidence a western Mediterranean cyclogenesis triggered by the passage of a deep MSLP minimum over central Europe. In the latter three cases the presence of a cyclone in the western MR is associated to a eastward pressure gradient and a high pressure over the eastern MR. Cyclones producing positive SLAs in the Levantine basin (at Iskenderun, Alexandria and Tripoli) move along the south-eastern part of the Mediterranean storm track (see figure 1 of Lionello et al (2016)) after having been generated in the north western part of the basin. The evolution of a cyclone moving along the north-western Mediterranean storm track is also evident for negative SLAs in the north-western MR (Alicante, Toulon, Trieste). For these three stations, the presence of a cyclone in the eastern MR is associated to a westward pressure gradient and a high pressure in the western MR. For negative SLAS at Iskenderun and Alexandria in the eastern MR the cyclone centres remains in the western MR, not far from cyclogenetic areas close to the Iberian peninsula, with a eastward pressure gradient associated with a high pressure in the eastern MR. Composites of negative SLAs at Dubrovnik Tripoli and Thessaloniki show a persistent high pressure over a large part of the basin during the event without the presence of a cyclone inside the MR. For negative SLAs in Gabes (at to a lower extent in Trieste) the position of the cyclone is such to produce a strong offshore wind blowing over shallow water areas.

### 3.3 Cyclone tracks associated to large sea level anomalies

To assess the link between the presence of a cyclone and large SLAs, the procedure adopted by Reale and Lionello (2013) for associating cyclones and precipitation extremes has been adapted to storm surges and applied to the ERA-Interim MSLP fields. The procedure consists of a sequence of steps: 1) a MSLP minimum is searched within a radius of 20 degs from the considered coastal station in the 6-hourly field, whose time is closest to the time of maximum amplitude of each positive and negative SLA event; 2) if only one minimum is found the corresponding cyclone is associated to the event; 3) if more than one minimum is present inside the search radius, the closest minimum is selected. When no minimum is found the event is not assigned to any cyclone.

30      Figure 6 shows the density tracks of cyclones associated with large positive SLAs and the positions of the cyclone centres at the peak of the events (blue squares). Density is computed splitting the domain in cells of 1.5 degs size and counting the cyclone centres located within each cell as each cyclone moves along its track. Units are probability per square kilometre. Each panel refers to a different coastal location (red square). It is evident that cyclone tracks initiate in the well-known cyclogenetic areas located inside and around the Mediterranean Sea (north Africa, south of the Atlas Mountains, Gulf of Genoa, Adriatic

Sea, Cyprus Area, Black Sea area) and in the North-East Atlantic (H.M.S.O. (1962); Alpert et al. (1990); Trigo et al. (1999); Maheras et al. (2001); Trigo et al. (2002); Lionello et al. (2002); Garcies and Homar (2011); Nissen et al. (2010); Campins et al. (2011); Reale and Lionello (2013); Lionello et al (2016)). The prevalent motion of the cyclone center is south-eastwards along the Mediterranean branch of the Mediterranean storm track. The importance of Atlantic cyclones decreases as the coastal

station location moves eastwards, while, simultaneously, the importance of cyclogenesis over the Mediterranean Sea increases, shifting towards the Levantine basin for the most south-eastern stations (Iskenderun and Alexandria). Cyclones originated over the North Africa are particularly relevant for Tripoli and Gabes and join the Mediterranean storm track in the eastern Mediterranean. In most cases, cyclone centres at the time of the SLAs are located so that the associated flow is directed towards the coast. In fact, the wind contributes a fraction of SLA in all location, but it is dominant factor only in locations that are char-

acterized by a long shallow water fetch, mainly Trieste and Gabes (Conte and Lionello (2013)), producing, depending on its direction with respect to the shore, a set-up or set-down effect.

Figure 7 shows the track density and cyclone positions for negative SLAs and relevant differences with respect to positive SLAs. In some stations (Toulon, Trieste, Tripoli and Gabes) the cyclones centres are concentrated in positions that determine

a wind blowing offshore at the time of the negative SLAs (wind set-down), while in other stations the cyclone centres are broadly distributed with several accumulation areas. Note that, at the time of the negative SLA event, for Trieste and Gabes, the cyclones are close to the station, while in other cases (such as Toulon, Alicante, Tripoli, Dubrovnik, Thessaloniki) they are positioned over the part of the Mediterranean basin opposite to that where the event occurs. In the two former stations the negative SLA is caused by the wind set-down. In the latter stations, the location of the cyclone is consistent with the prevalence

of high pressure conditions in the location where negative SLAs occur, which is related to the cyclone center through a strong MSLP gradient across the basin (see also section 3.5).

Cyclone track densities in the panels of figures 6 and 7 evidence the Mediterranean storm track, along which most cyclones move after having been originated inside the Mediterranean. Differences are the signature of cyclones entering the Mediterranean Sea from North Africa (North-East Atlantic) that are the stronger (weaker) in figure 6 than in 7. Comparing figures 6

and 7 we infer that the same cyclone can eventually be associated to positive and negative SLAs in different parts of basin. For example, the tracks of cyclones associated with negative SLAs in Alicante, Toulon and Trieste are similar to those associated with positive SLAs in Thessaloniki, showing that the same cyclone when moving along the main branch of the Mediterranean storm track can produce negative SLAs at the former stations and positive at the latter.

**3.4   Statistics of cyclone center positions during large sea level anomalies**

In this subsection we investigate to which extent the position of the cyclone centres at the time of the positive and negative SLAs (figures 6 and 7) are statistically linked to the events themselves. This is analysed by computing the probability (relative frequency) that a cyclone is present within a 10deg radius from the reference position, which is denoted with the yellow square, and comparing it to the background "climatological" probability. The latter is estimated searching for cyclone centres in a se-

quence of MSLP fields extracted with a time step of 10 days (to avoid correlations among successive fields). If the difference between the two values is significant (the t-test of the mean with a significant level of 95% has been used), then the presence of the SLA is statistically linked to the position of the cyclone in an area surrounding the reference position. The reference position is the center of the 5deg wide lat-lon cell where the density of cyclone centres (blue squares in figures 6 and 7) has a

maximum. This procedure is repeated for each station. For negative SLAs at Iskenderun, where this criterion would locate the reference at the eastern boundary of the map, the second largest maximum (in the middle of the Ionian Sea) has been used.

Further, for each cyclone associated to positive and negative SLAs the cyclogenesis point (where its trajectory starts) has been identified and attributed to four areas: Atlantic (Atl), Northern Africa (Afr), Western Mediterranean (WM), Eastern Mediter-

ranean (EM) and Asia-Europe (AsEu), according to the main cyclogenetic areas that are suggested by the existing literature (e.g. Lionello et al (2016) and figure 1).

Table 2 compares the probability that a cyclone is around the reference position for positive SLA (column $P_{SLA+}$) with the climatological probability (column $P_{clim}$). Each line denotes a different station. At all stations, differences are statistically

significant at the 95% level. Results show that for all stations the probability to find a cyclone around the reference position in coincidence with a positive SLA is higher than the climatological mean.

Further, table 2 reports the relative frequency of cyclogenesis in the four different areas. The most important cyclogenesis areas is the WM for all stations, except for Iskenderun and Alexandria, where it is the Em, and Tripoli, where it is Afr. Afr

is a important cyclogenetic area also for Gabes (and it plays an important role also for Alexandria and Iskenderun). Atl is an important source of cyclones only for stations located in the western Mediterranean (Alicante, Toulon, Trieste). The tracking algorithm fails to find a cyclone within a 10deg radius from the reference position with comparatively large frequency in Gabes (37%), Toulon (29%), Alicante (28%) and Trieste (25%).

The link with cyclones is statistically significant for all stations also for large negative SLAs (table 3). However, the percentage of "not assigned" events is generally higher than for positive SLAs, and it is particularly large for Dubrovnik and Tripoli, where almost 50% of large negative SLAs occurred when no cyclone was present within a 10deg radius from the reference position. This is actually consistent with the MSLP composites in figure 3.

The cyclogenesis areas that have been identified for positive SLA are confirmed, but for negative SLAs their relative importance is different (table 3). With respect to positive SLAs, the frequency of Atl cyclogenesis is more important for negative SLAs in the Eastern than in the Western Mediterranean. Considering large negative SLAs, the WM keeps a leading role, being the main overall source and individually for most stations (Alicante, Toulon, Trieste, Dubrovnik, Tripoli and Gabes), while very few cyclones are generated in the EM, and Afr is the main cyclogenesis area for Alexandria and Iskenderun.

| Location | $P_{SLA+}$ | $P_{clim}$ | Atl | Afr | Wm | Em | AsEu | not assigned |
|---|---|---|---|---|---|---|---|---|
| Alicante | 72 | 41 | 22 | 23 | 27 | 0 | 0 | 28 |
| Toulon | 71 | 41 | 27 | 6 | 37 | 9 | 1 | 29 |
| Trieste | 75 | 41 | 28 | 9 | 28 | 0 | 10 | 25 |
| Dubrovnik | 90 | 41 | 13 | 8 | 62 | 1 | 6 | 10 |
| Thessaloniki | 93 | 42 | 13 | 8 | 57 | 8 | 7 | 7 |
| Iskenderun | 81 | 32 | 5 | 10 | 21 | 36 | 9 | 19 |
| Alexandria | 83 | 32 | 4 | 22 | 24 | 32 | 3 | 15 |
| Tripoli | 96 | 41 | 8 | 30 | 48 | 8 | 2 | 4 |
| Gabes | 63 | 45 | 3 | 56 | 4 | 0 | 0 | 37 |

**Table 2.** Statistics of cyclones producing the 100 largest positive sea level anomalies in each considered station. The first two columns, labelled "$P_{SLA+}$" and "$P_{clim}$, report the probability (%) to find a cyclone within a 10degs search radius from the reference point (denoted with a yellow square in figure 6) at the time of the event and the corresponding climatological mean value, respectively. Differences between the "$P_{SLA+}$" and "$P_{clim}$" values are statistically significant at the 95% level. Following columns (labelled "Atl","Afr","Wm","EM","AsEu") report probability (%) that the cyclogenesis of the cyclone associated with the event occurred in the areas shown in figure 1. The last column reports the number of events in the period 1979-2001 that were not assigned to any cyclone.

Figure 8 shows the centrers of anticyclones at the time of negative SLAs. It is made following the same procedure that has been used for figure 7, which refers to cyclones. It reveals the location of centres of anticyclones in the areas where the right column of figure 5 shows high pressure systems. Anticyclones are actually concentrated around the stations, with the exception of Gabes and, to a lesser degree, Trieste, where the wind effect is much larger than the inverse barometer effect and anticyclones play a minor role. Therefore, negative SLAs are linked to the presence of a high pressure around the station. This is necessarily true for most stations, because of the inverse barometer effect. However (see table 4), the probability to find an anticyclone at a distance lower than 10 degrees from the reference position[3] at the time of negative SLAs is significantly larger than the climatological value only for three stations (Toulon, Thessaloniki, Iskenderun). On the contrary, in Gabes, the absence of an anticyclone is linked to negative SLAs (and this is justified by the dominant role of the wind at this station). The link with the presence of a cyclone in the part of the basin opposite to the station (table 3) is stronger than what shown in table 4 for anticyclones.

---

[3]The reference position is defined as the center of the 5deg wide lat-lon cell where the density of anticyclone centres (blue square in figure 8) has a maximum (same procedure that was adopted for table 2)

| Location | $P_{SLA-}$ | $P_{clim}$ | Atl | Afr | Wm | Em | AsEu | not assigned |
|---|---|---|---|---|---|---|---|---|
| Alicante | 92 | 42 | 12 | 16 | 48 | 7 | 9 | 8 |
| Toulon | 68 | 41 | 7 | 16 | 36 | 6 | 3 | 22 |
| Trieste | 75 | 42 | 7 | 9 | 47 | 7 | 5 | 25 |
| Dubrovnik | 54 | 42 | 14 | 19 | 20 | 0 | 1 | 46 |
| Thessaloniki | 67 | 38 | 11 | 43 | 13 | 0 | 0 | 33 |
| Iskenderun | 71 | 41 | 15 | 31 | 21 | 1 | 3 | 29 |
| Alexandria | 76 | 45 | 14 | 39 | 22 | 1 | 0 | 24 |
| Tripoli | 51 | 42 | 11 | 13 | 26 | 0 | 1 | 49 |
| Gabes | 82 | 41 | 17 | 23 | 37 | 1 | 4 | 18 |

**Table 3.** same as table 2 except it refers to negative sea level anomalies.

| Location | $P^A_{SLA-}$ | $P^A_{clim}$ | not assigned |
|---|---|---|---|
| Alicante | 38 | 40 | 62 |
| Toulon | **56** | 40 | 44 |
| Trieste | 44 | 37 | 56 |
| Dubrovnik | 42 | 37 | 58 |
| Thessaloniki | **62** | 41 | 38 |
| Iskenderun | **51** | 31 | 49 |
| Alexandria | 43 | 39 | 57 |
| Tripoli | **33** | 44 | 67 |
| Gabes | 40 | 48 | 60 |

**Table 4.** Same as table 2 except that it refers to negative sea level anomalies, to the presence of anticyclones, and the columns with information on the genesis of the anticyclones are not reported. The columns, labelled "$P^A_{SLA-}$" and "$P^A_{clim}$, report the probability (%) to find an anticyclone within a 10degs search radius from the reference point (denoted with a yellow square in figure 8) at the time of the event, the corresponding climatological mean value, and to fail detecting the presence of an anticyclone, in this order. Bold values denote differences between $P^A_{SLA-}$ and $P^A_{clim}$ that are statistically signicant at the 95% level.

## 3.5 The link of the SLA values with cyclone positions and intensities

Here we investigate the SLAs at the 9 considered stations as function of the position of cyclones inside the MR. Each panel in figure 9 considers a different coastal station and shows the mean SLA at the selected station as function of the position of the cyclone centre. Each panel has been obtained dividing the MR in 1.5x1.5 degs lat-lon cells, selecting the time when a cyclone center is within each cell, extracting from the time series the corresponding SLA values at the considered station, computing their average and attributing it to the cell. Note that these average SLAs result from cyclones of different intensity and structure, some of them eventually very shallow.

Figure 9 allows to distinguish between local and remote effects associated with the passage of cyclones. The inverse barometer effect caused by the low pressure values around the centre of the cyclones is shown by the positives SLA values in the area surrounding the stations (which is particularly intense in Trieste, Dubrovnik and Thessaloniki). The wind set-up is shown by positive (negative) values in the areas where the cyclone centre would cause a wind pushing water masses toward (away from) the coastal station. In general, the shift of the areas with positive/negative values with respect to the station is the evidence of the importance of the wind set-up component, which is large in the presence of a shallow water fetch. The role of the wind set-up is particularly evident in Gabes (figure 9i), where cyclones located above North Africa (west of the Sicily Strait) produce positive (negative) SLAs, and the inverse barometer effect is negligible in comparison. The situation in Tripoli is analogue to Gabes.

In almost all panels of figure 9 significant negative values are present in areas of the basin relatively far away from the stations, at a distance that is too large for a direct action of the wind and low pressure of cyclones on the sea level. Two examples are the westernmost (Alicante) and easternmost stations (and Alexandria), where negative SLAs are associated to a cyclone above the central Mediterranean. In these cases, the presence of a cyclone is associated to a pressure gradient across the basin (see also figure 10), producing a redistribution of mass and a temporary decrease of sea level in areas far away from the cyclone center. However, in the panels of figure 9, positive features are larger than negative features, meaning that the association between SLAs and cyclone positions is larger for positive than for negative events.

The link between MSLP anomaly at the coastal station and position of the cyclone is shown in figure 10. This figure has been obtained following exactly the same procedure as figure 9, except it plots the difference of MSLP between the station and the cyclone center (and not the SLA). The small values around the position of each coastal station are the obvious consequence of the cyclone center being close to the station. The positive values, between 10 and 15hPa in the areas of the basin opposite to the station evidence that, when the cyclone center is located in such areas, the pressure at the station is high and the inverse barometer effect contributes to negative sea level anomalies.This happens at Alicante, Toulon, Thessaloniki, Trieste when a cyclone is located in areas where the density of cyclone centre in figure 7 is large. This explains, on the basis of the inverse barometer effect, the link between negative SLAs and the presence of a cyclone at a distance that is too large for its direct

action on the sea level at the considered station, though other factors are involved (see section 3.6).

The dependence of the SLA value on the cyclone MSLP central minimum is analysed by computing the linear regression coefficient between the SLAs and the values of the MSLP minima. Each panel of figure 11 considers a different coastal station and adopts a procedure in part similar to figure 9. The whole MR has been divided in 1.5x1.5degs lat-lon cells. When a cyclone center is within a cell, the pair MSLP minimum and corresponding SLA anomaly at the coastal station is selected. For each cell the regression coefficient of SLA values with respect to MSLP minima has been estimated and plotted in figure 11 as a function of the cyclone centre position. Therefore, each panel of figure 11 is meant to show the sensitivity of the SLA at the considered station on the cyclone central MSLP minimum as a function of the cyclone position. Only values above the 90% confidence level are shown. Since SLA increases for decreasing value of pressure minima, in figure 11 the orange-red colors denote negative regression coefficient values (this is meant to facilitate the comparison with figure 9).

The inverse barometer effect would produce a value of -1cm/hPa when a cyclone center is exactly above the station. However, the regression coefficients do not reflect directly only the inverse barometer effect, which is a local relation, but also the contribution of wind or other dynamics to the SLAs. When the cyclone center is located away from the station the contribution of the inverse barometer effect decreases, because of the decreasing amplitude of the MSLP anomaly with the distance from the cyclone center. On the contrary, the importance of the contribution due to the wind set up might increase, if the position of the cyclone center establishes an atmospheric circulation accumulating water masses against the coastal station (or subtracting them from it). Trieste (figure 11) clearly shows that the dependence of the positive SLA on the intensity of the cyclones is largest when their centres are located in the gulf of Genoa, in the position such that the associate wind blowing along the Adriatic Sea will accumulates water at its northern shore.

In fact, the regression coefficients link MSLP minima and SLA values in locations that are separated by several hundreds of kilometres. The largest statistically significant values of regression coefficients have been found at a distance of approximately 700 km from the station for Trieste (-2.2 cm $hPa^{-1}$), Gabes(-1.7cm $hPa^{-1}$), Thessaloniki (-1.6cm $hPa^{-1}$), and Iskenderun (-1.5 cm $hPa^{-1}$). Values at other stations are smaller in module (less than -1.2 cm $hPa^{-1}$). However, these regression coefficients cannot be used for estimating extreme SLAs, when the impact of the atmospheric forcing is amplified by peculiarity of the atmospheric pattern.

Figure 11 provides a statistical evidence of a teleconnection[4] linking negative SLAs to cyclones whose centers are located thousands of kilometers far away from the station. The yellow areas far away from the stations in figure 10 show that a cyclone positioned in those areas is linked to negative SLAs at these two stations. Therefore, this figure clearly shows the connection between intensity of cyclones in the opposite part of the basin and negative anomalies at Iskenderun and Alexandria.

---

[4]Here teleconnection does not refer to a mechanism acting at global scale, but across different areas of the Mediterranean region

The patterns in figure 9 and 11 are consistent with the geographical distribution of cyclone centres in figures 6 and 7. The correspondence is not exact, but this should be expected. In fact, figures 6 and 7 include only the largest SLA events, while figure 9 represents the mean SLA response to all cyclones that crossed the MR during the hindcast.

## 3.6 Dynamics of sea level anomalies in the Mediterranean Sea

Figures 12 and 13 show the features of the SLAs over the whole Mediterranean Sea. Each panel is a composite of the fields corresponding to the 100 largest SLA maxima at each of the 9 coastal stations considered in this study. Figure 12 and 13 consider positive and negative SLAs, respectively. The four columns show the total SLA (first column, cm), the contribution due to the inverse barometer effect (second column, %), the residual (third column, %), meaning the difference between the first and the second column, which represents the part of SLA that is not explained by the inverse barometer effect. The fourth column shows the composites of the wind fields at the time of the SLAs largest amplitude. The maps displaying the inverse barometer and residual contributions show normalized values, obtained dividing by the SLA maximum (minimum) in the total positive (negative) corresponding SLA map (in other words, units are percentages of overall maxima and minima). The inverse barometer contribution is built using the 6-hourly MSLP of the COSMO-ERA dataset closest to the time when the SLA has the largest magnitude at the coastal station.

The presence of a cyclone at the time of the positive SLA event is shown by the shape of the SLA associated to the inverse barometer effect, which mimics the contour lines around a cyclone center. In the case of positive SLA (figure 12) the distance between the station and the maximum of the inverse barometer effect (central column) is an indication of the importance of the wind (particularly large for Gabes), which is made evident by a local and large SLA in the area around to the station in the maps of the residual (right column).

The presence of a cyclone is evident also in the composites of negative SLAs (figure 13 , values are negative because they are normalized using the SLA minimum), with the exception of Dubrovnik. However, in these cases, the cyclone centres are located in the opposite part of the basin with respect to the position of the stations. Several different dynamics are suggested by these maps. At Tripoli, Alexandria, Iskenderun, Thessaloniki the presence of a cyclone is associated to a pressure gradient across the basin such that the inverse barometer effect contributes substantially to the negative SLA in the position of the station. For Gabes and Trieste, where there is a long shallow fetch, the atmospheric circulation around the cyclone center produces a set-down at the station, which is shown by the locally large contribution to the negative SLA in the area around to the station in the maps of the residual (right column).

Some stations show a large residual that, because of its extension, cannot be associated to the action of the wind and it reflects a contrast of level between the western and eastern parts of the Mediterranean Sea. In some cases, such as Alicante, Toulon, Iskenderun and Tripoli, it produces a very substantial contribution to the magnitude of the SLA, particularly when it

is negative. This contribution amplifies the contrast of sea level between western and eastern part of the Mediterranean that is already produced by the difference of MSLP associated with the presence of a cyclone inside the basin. This suggests that, in the model simulations, during the synoptic system evolution, the SLA has no sufficient time to reach equilibrium with the MSLP field and the Gibraltar strait does not allow sufficient water flow to comply with the inverse barometer effect. In practice, during the development of both positive and negative SLAs at the coastal stations, the average SLA of the whole Mediterranean changes little, in spite of the forcing caused by the inverse barometer effect. Whether this is realistic or an artificial model feature remains to be investigated.

The action of the wind is evident in the fourth column of figures 12 and 13, which show the composites of the wind fields at the time of the SLAs largest amplitude. In these maps, the presence of a strong wind blowing towards the coast (figure 12, positive SLAs) or offshore (figure 13, negative SLAs) is consistent with the large residuals at Trieste, Tripoli and Gabes. For positive SLAs the wind is also present in correspondence with the residuals (which are smaller than in the previous stations) at Alexandria, Iskenderun and Thessaloniki.

## 4 Discussion and conclusions

This study, to the best of our knowledge, represents the first analysis of the synoptic conditions leading to positive/negative SLAs along the coasts of the entire Mediterranean Sea. An SLA hindcast has allowed to describe the link between cyclones and positive/negative SLAs. Nine coastal stations distributed along the entire Mediterranean coastline are considered and meant to be representative of the variety of conditions leading to large SLAs.

The presence of cyclones and their different role in relation to the position of the considered station is evident in MSLP composites (Figures 4 and 5 ), which show the different synoptic evolutions leading to large SLA along the different parts of the Mediterranean coastline. The association of SLAs with cyclones is shown also by the spatial distributions of the cyclone centers in figures 6 and 7. A statistical estimate of the significance of the link between cyclones and SLAs is found in figures 9 and 11 and further supported by tables 2 and 3. Figure 9 links the presence of a cyclone to the SLAs. Figure 11 links the level of the SLAs to the intensity of the cyclone. Figure 9 shows that the presence of a cyclone in areas closely matching those of low pressure centers in the composites in figures 6 and 7 is associated to positive and negative SLAs.

The pattern associating the level of the negative SLAs to the value of the pressure minimum (figure 11) is weaker than that in figure 9. This is because the level of positive SLAs is caused by the direct local action of the cyclone. The level of negative SLAs is caused mostly by the direct action of the wind around the cyclone centre in Gabes and Trieste. In other stations it is caused by the pressure gradient across the basin, often correlated to the presence of a cyclone center in the opposite part of the basin (figure 9), and by redistribution of water mass within the different parts of the basin (figure 13), which is not explained by a local inverse barometer effect. The link between large negative SLAs and cyclones is a statistical concept. It is not meant that

cyclones are the direct cause of large negative anomalies, but an indirect driver of them, because the synoptic condition leading to negative anomalies is frequently associated to the present of a cyclone in the opposite part of the basin. Only at Dubrovnik, and to a lesser extent at Tripoli, the value of negative SLAs is dominated by the local high pressure, with the inverse barometer effect and the action of the wind being the dominant factor in the former and the latter, respectively.

We clarify that, obviously, we are not denying that high pressure leads to a negative sea level. Our study clearly supports the importance of the local action of the inverse barometer effect for both positive and negative SLAs. The link between large negative SLAs and cyclones that is shown in this study does not describe a local effect, but a synoptic scale teleconnection, supported by a statistical analysis and explained by the large scale structure of the SLP fields. The connection between cyclone in the opposite part of the basin and negative SLAS at the station is mediated by the cross basin pressure gradient and the presence of a high pressure that locally acts according to the inverse barometer effect.

Cyclones characteristics (area of cyclogenesis, track of the central minimum and position at the time of the event) differ for each location and in function of SLA's sign. Atlantic cyclones are associated mainly with positive SLAs in the western basin while positive/negative SLA at the coasts of the south eastern/north western parts of the basin are associated with cyclones generated in the western basin. More specifically, for positive SLAs, the area where most cyclones causing them are generated is the Wm, except for Iskenderun and Alexandria, where it is the Em, and Tripoli, where it is Afr, which is an important cyclogenetic area also for Gabes, and to a lesser extent for Alexandria and Iskenderun. For large negative SLAs, Wm is main overall source, being the largest individually for most stations (Alicante, Toulon, Trieste, Dubrovnik, Tripoli and Gabes), while very few cyclones are generated in the EM, and Afr is the main cyclogenesis area for Alexandria and Iskenderun.

The internal variability of the sample is explored considering the area of cyclogenesis, which allows to distinguish the different evolutions of cyclones. We have adopted this process-oriented approach. In our opinion, it is very plausible that PCA or clustering of the trajectories would have produced very similar outcomes. We admit that our approach might hide some aspects of the internal variability of the sample related to different synoptic patterns at the time of the SLAs, which might be worth to explore in a future studies for those stations where this issue would eventually result significant.

The hindcast allows to compare the relative importance of three factors causing large SLAs: the inverse barometer effect, the contribution caused by the action of the wind and a residual water mass redistribution within the basin. The inverse barometer effect is mainly associated to the presence of a MSLP minimum/maximum in an area surrounding the station and, for negative SLAs, also the pressure gradient across the basin that is statistically associated to the presence of a cyclone center in the opposite part of the basin. The residual water mass redistribution is linked to the evolution of the MSLP. It is possibly explained by transient water mass unbalance within the Mediterranean Sea in the simulation of the water exchange with the Atlantic in the model hindcast. The action of the wind is evident and large in the shallow areas of the Mediterranean Sea (mainly in the

North Adriatic and Gulf of Gabes). Its magnitude is related to the intensity and shape of the atmospheric circulation around the central pressure minimum of the cyclone and, depending on its positions contributes to positive or negative SLAs.

*Competing interests.* No competing interests are present

*Acknowledgements.* Marco Reale has been supported in this work by OGS and CINECA under HPC-TRES program award number 2015-07.

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

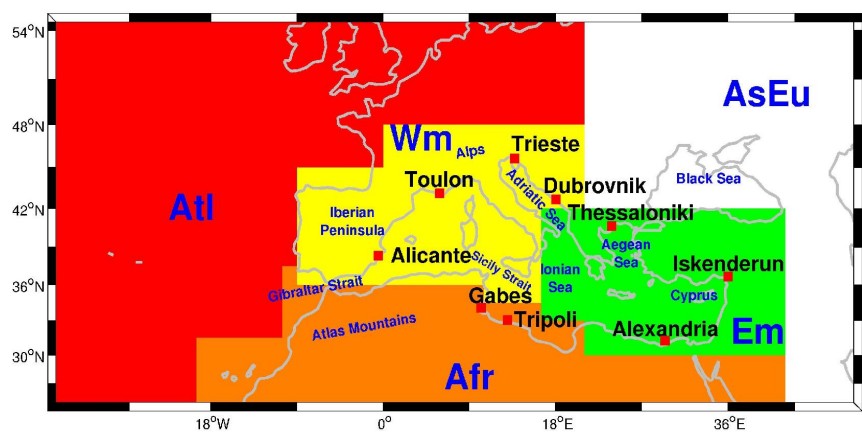

**Figure 1.** The Mediterranean region with the coastal stations considered in this study (starting from the west in clockwise direction along the coastline): Alicante, Toulon, Trieste, Dubrovnik, Thessaloniki, Iskenderun, Alexandria, Tripoli, Gabes. The figure shows also the areas considered for the cyclogeneses producing sea level anomalies in the Mediterranean region : Atlantic (Atl), Africa (Afr), Western Med (Wm), Eastern Med (Em), Asia and Europe (AsEu). Finally, the geographical names used in the text are annotated in the map.

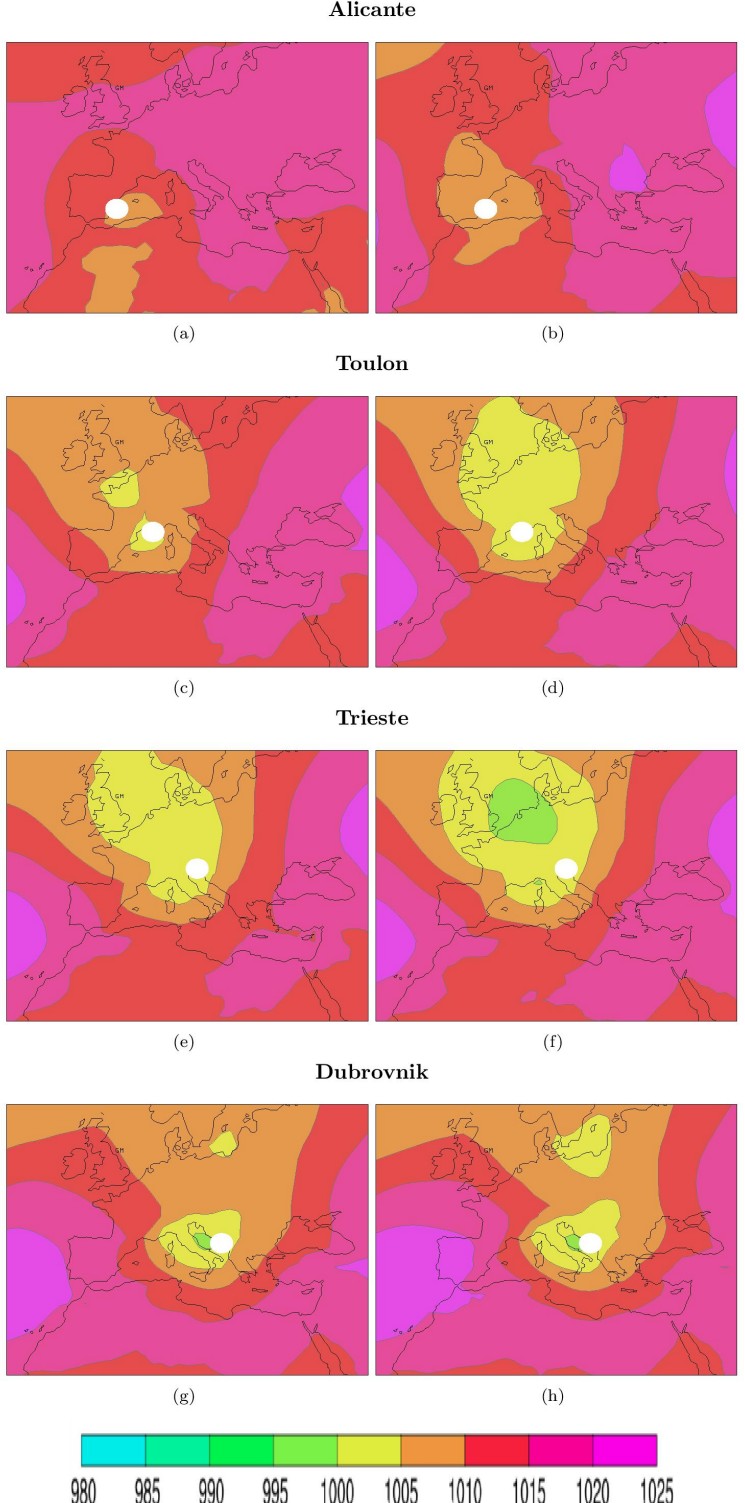

**Figure 2.** Composite of ERA-Interim SLP fields (values in hPa) associated with large positive sea level anomalies in Alicante (a,b), Toulon (c,d), Trieste (e, f) and Dubrovnik (g,h), denoted with white circles in the maps. The right and left column show composites based on the COSMO-ERA hindcast ("MOD") and in the observed time series ("OBS"), respectively. Each composite is based on a total number of 100 events, obtained selecting the largest events in the 1979-2001 period.

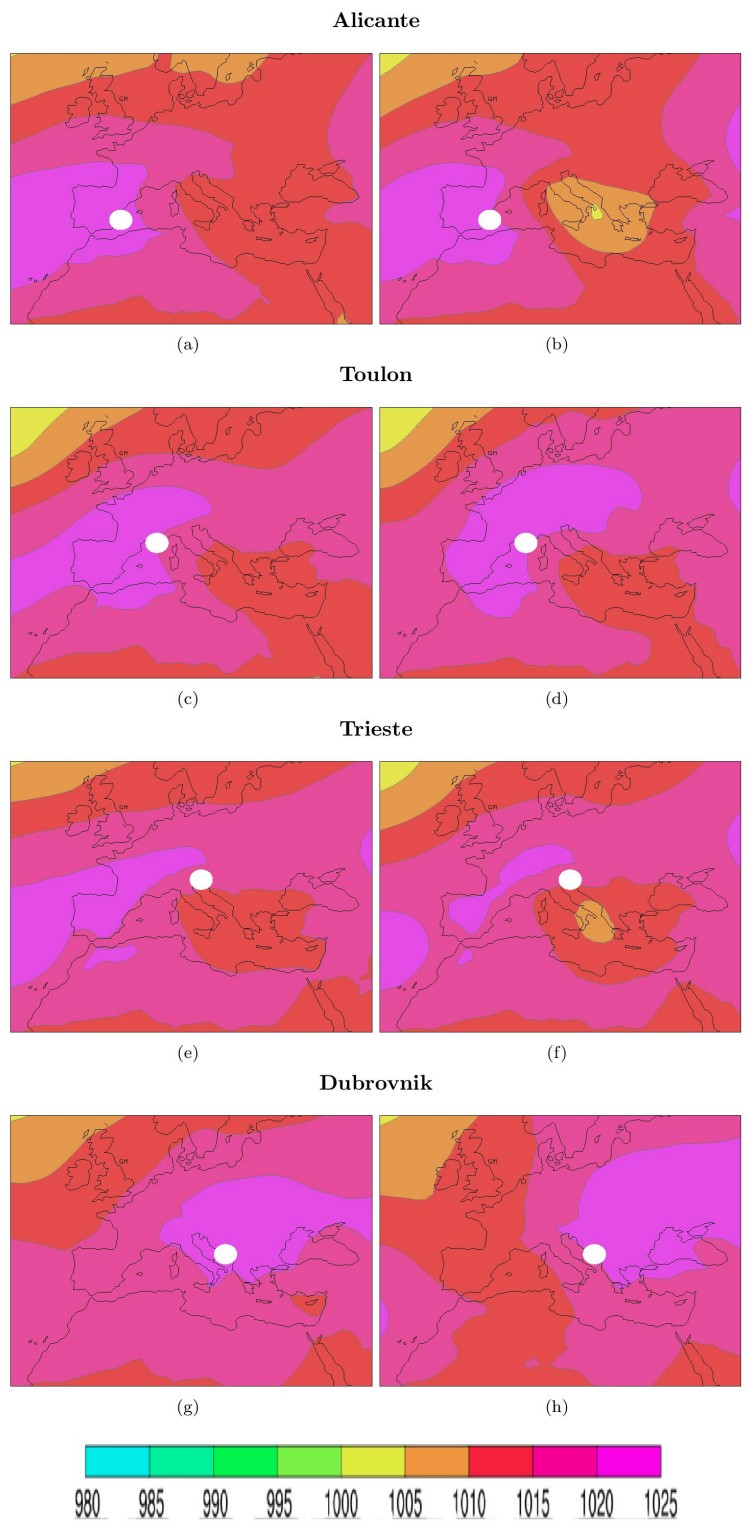

**Figure 3.** Same as figure 2, except it refers to large negative anomalies.

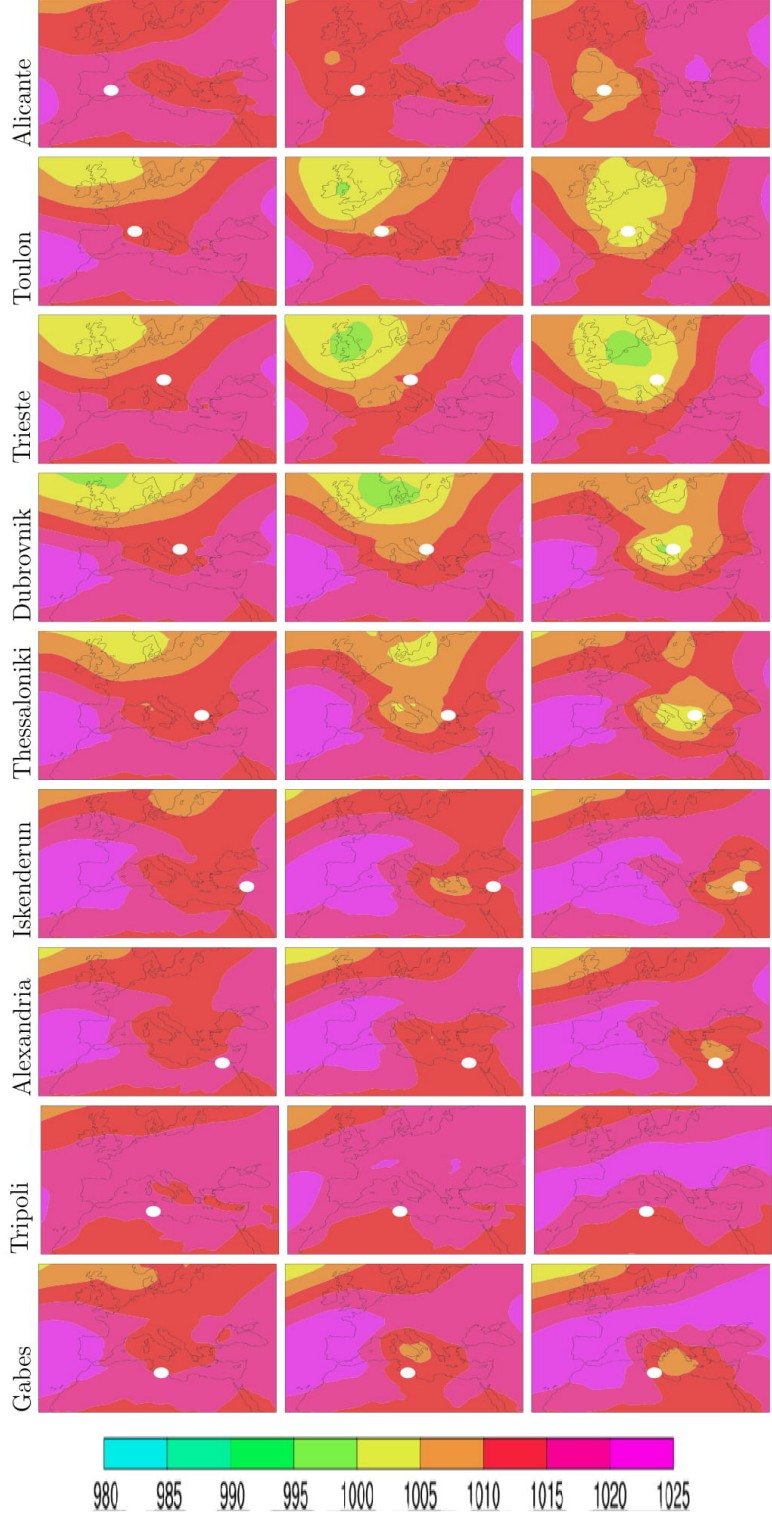

**Figure 4.** Composites of ERA-Interim MSLP fields (in hPa) associated with the values of large positive sea level anomalies 48h (left column), 24h (middle column) before and at the peak (right column) of the event in Alicante, Toulon, Trieste, Dubrovnik, Thessaloniki, Iskenendur, Alexandria, Tripoli, Gabes, denoted with white circles in the maps.

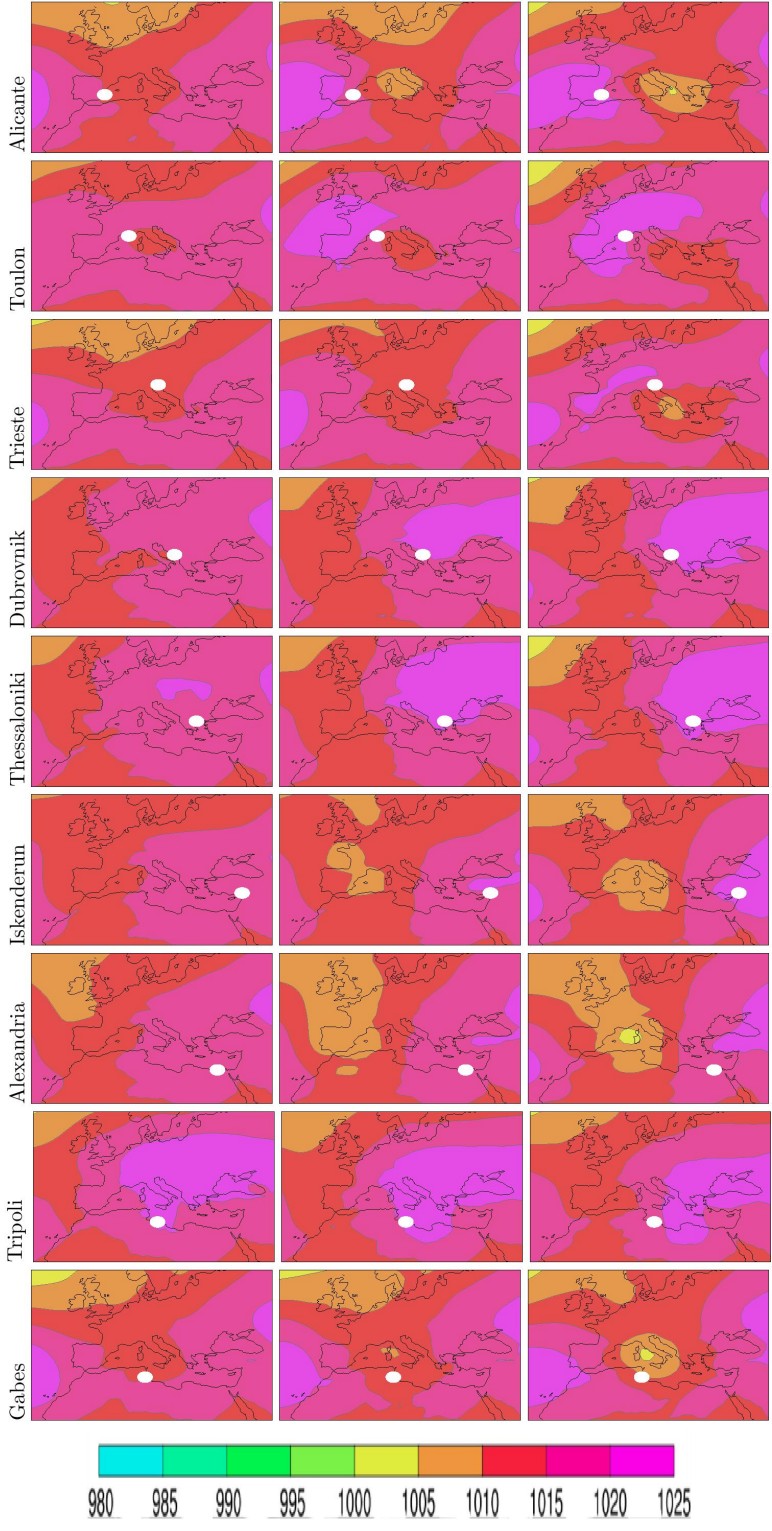

**Figure 5.** Same as figure 4, except it refers to large negative sea level anomalies.

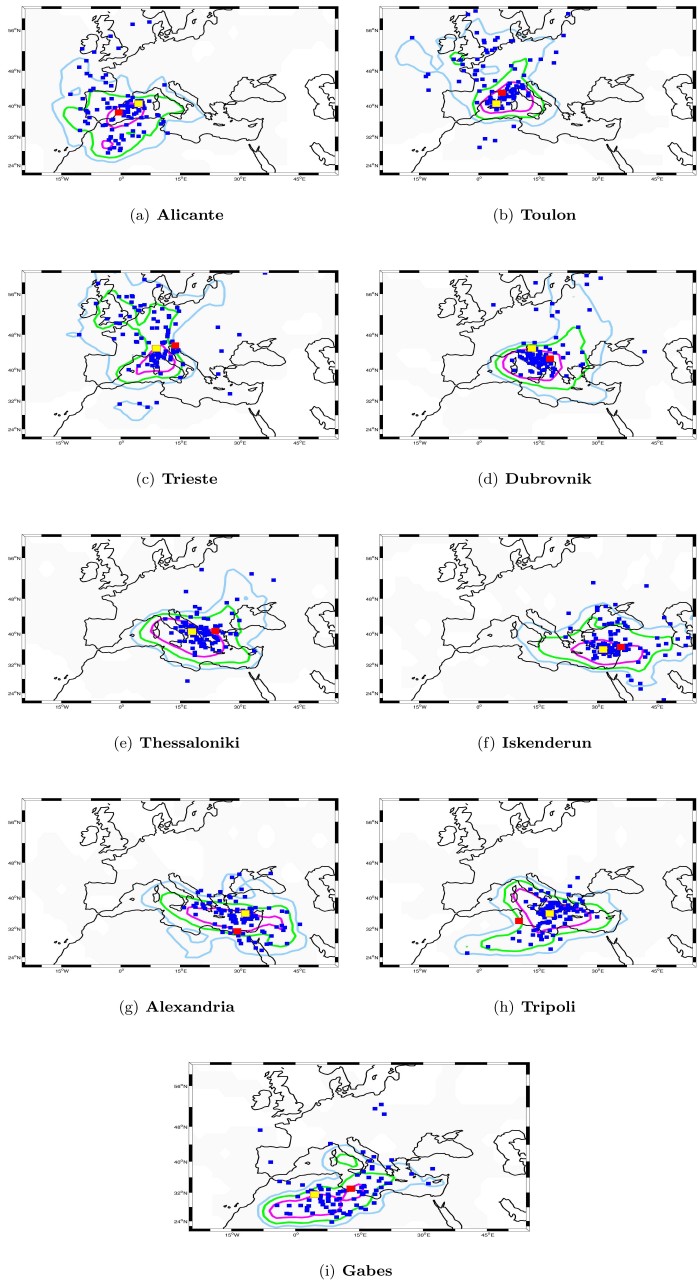

**Figure 6.** Track density of cyclones producing large sea level anomalies at Alicante (a), Toulon (b), Trieste (c), Dubrovnik (d), Thessaloniki (e), Iskenderun (f), Alexandria (g), Tripoli (h), Gabes (h) (locations are denoted with a red square). Blue squares show the position of the cyclone centres at the peak of the sea level anomaly. The yellow square denotes the reference position used in table 2 and subsection 3.4. A smoothing radius of 5degs is applied to the data original resolution (1.5degs). Contour lines are drawn at the $.25 \cdot 10^{-7}$ (blue line), $0.5 \cdot 10^{-7}$ (green line), $1 \cdot 10^{-7}$ (magenta line) levels. Units are probability per square kilometre

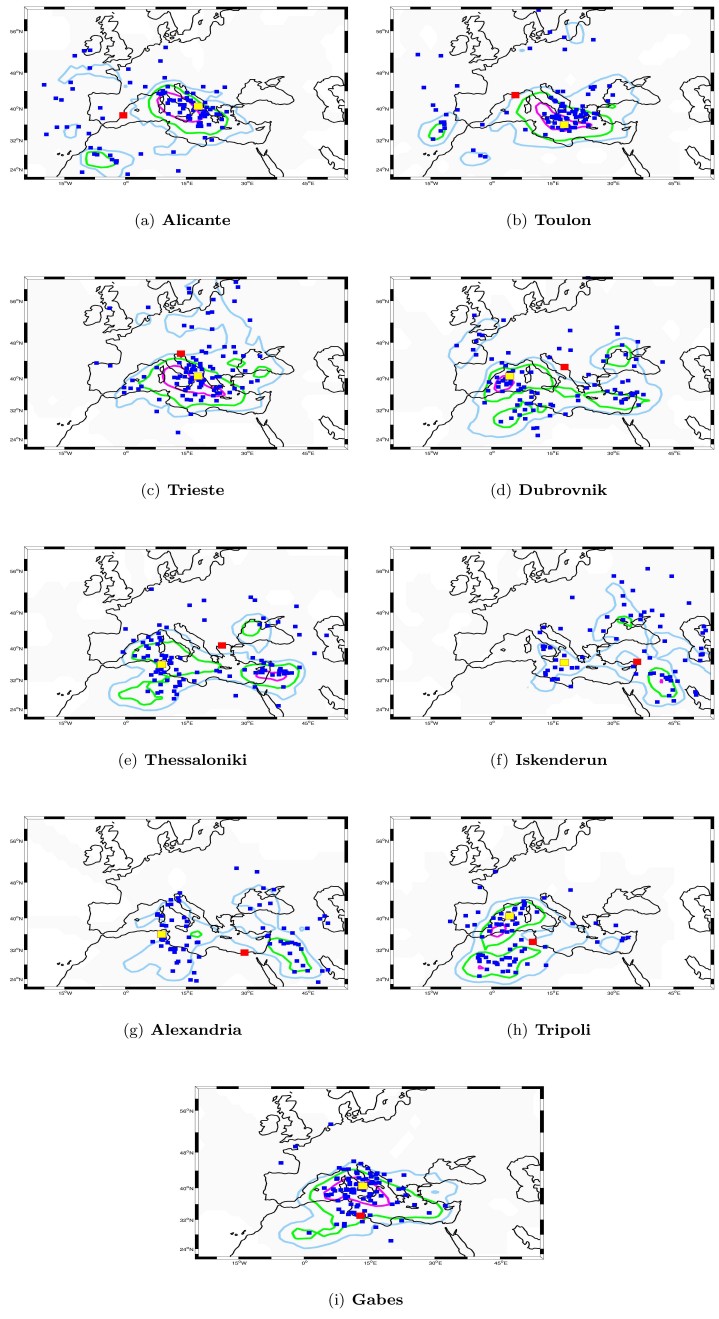

**Figure 7.** Same as figure 6, except it refers to large negative sea level anomalies and to table 3.

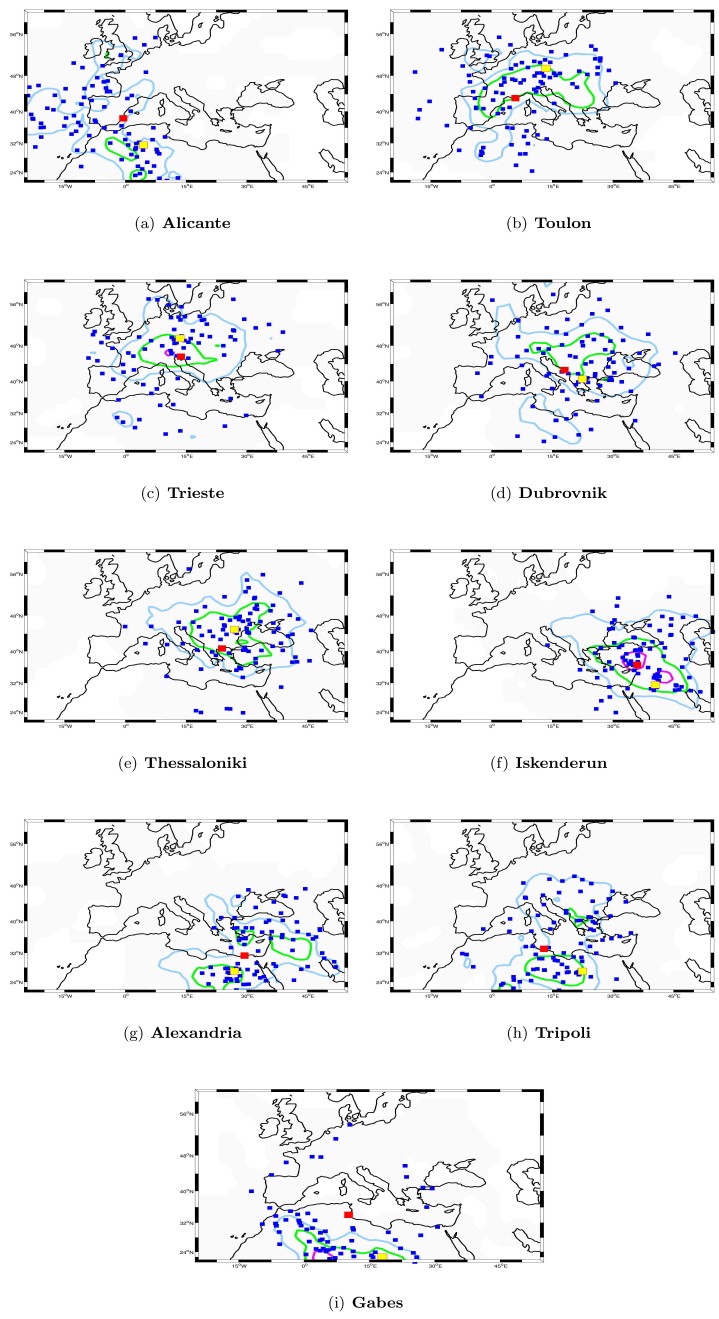

(a) **Alicante**

(b) **Toulon**

(c) **Trieste**

(d) **Dubrovnik**

(e) **Thessaloniki**

(f) **Iskenderun**

(g) **Alexandria**

(h) **Tripoli**

(i) **Gabes**

**Figure 8.** Same as figure 6, except it refers to large negative sea level anomalies, to the presence of anticyclones and to table 4.

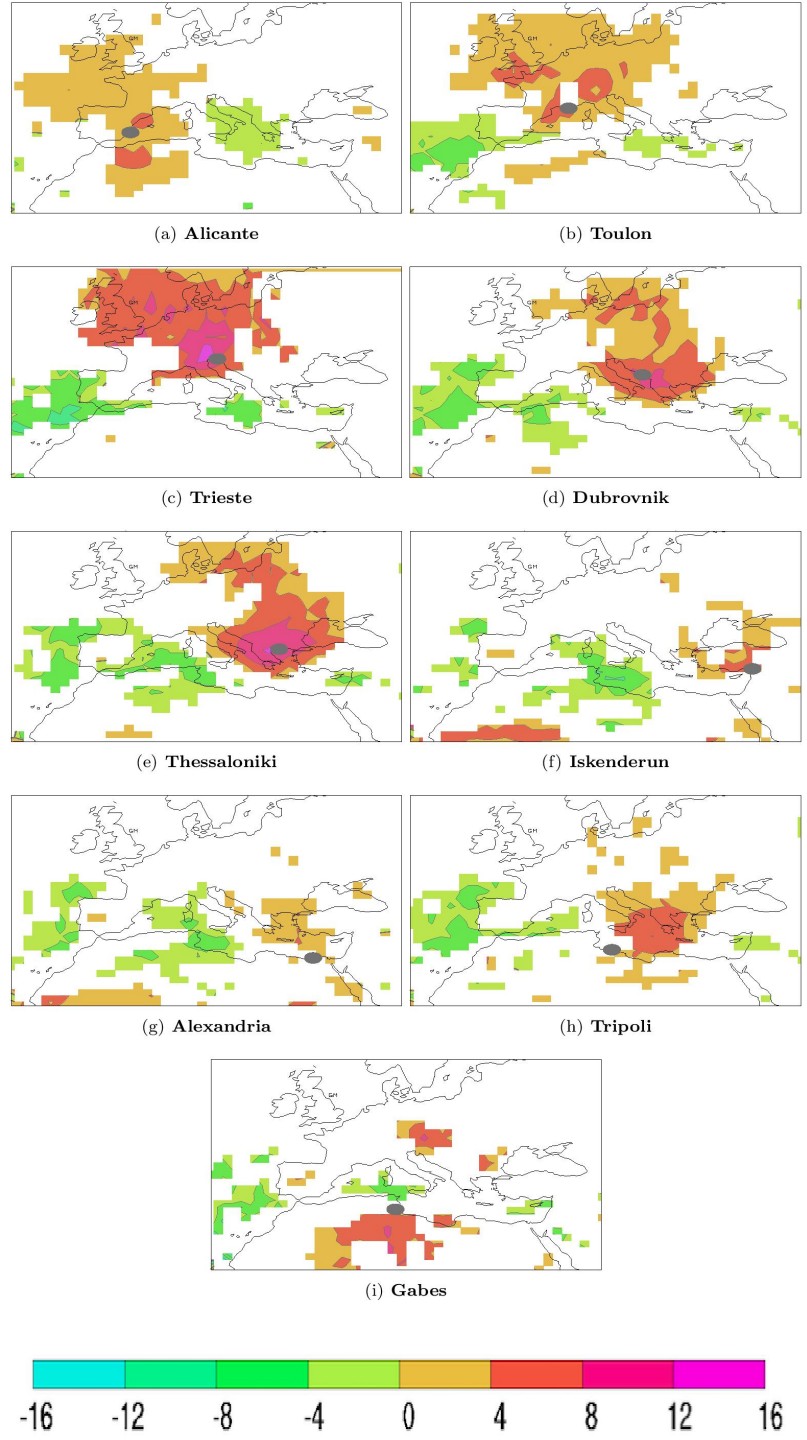

**Figure 9.** Mean SLA (cm) at the coastal station as function of cyclone positions. Each panel considers a different coastal station: Alicante (a), Toulon (b), Trieste (c), Dubrovnik (d), Thessaloniki (e), Iskenderun (f), Alexandria (g), Tripoli (h), Gabes (h). Only values reaching the 90% significance level are shown.

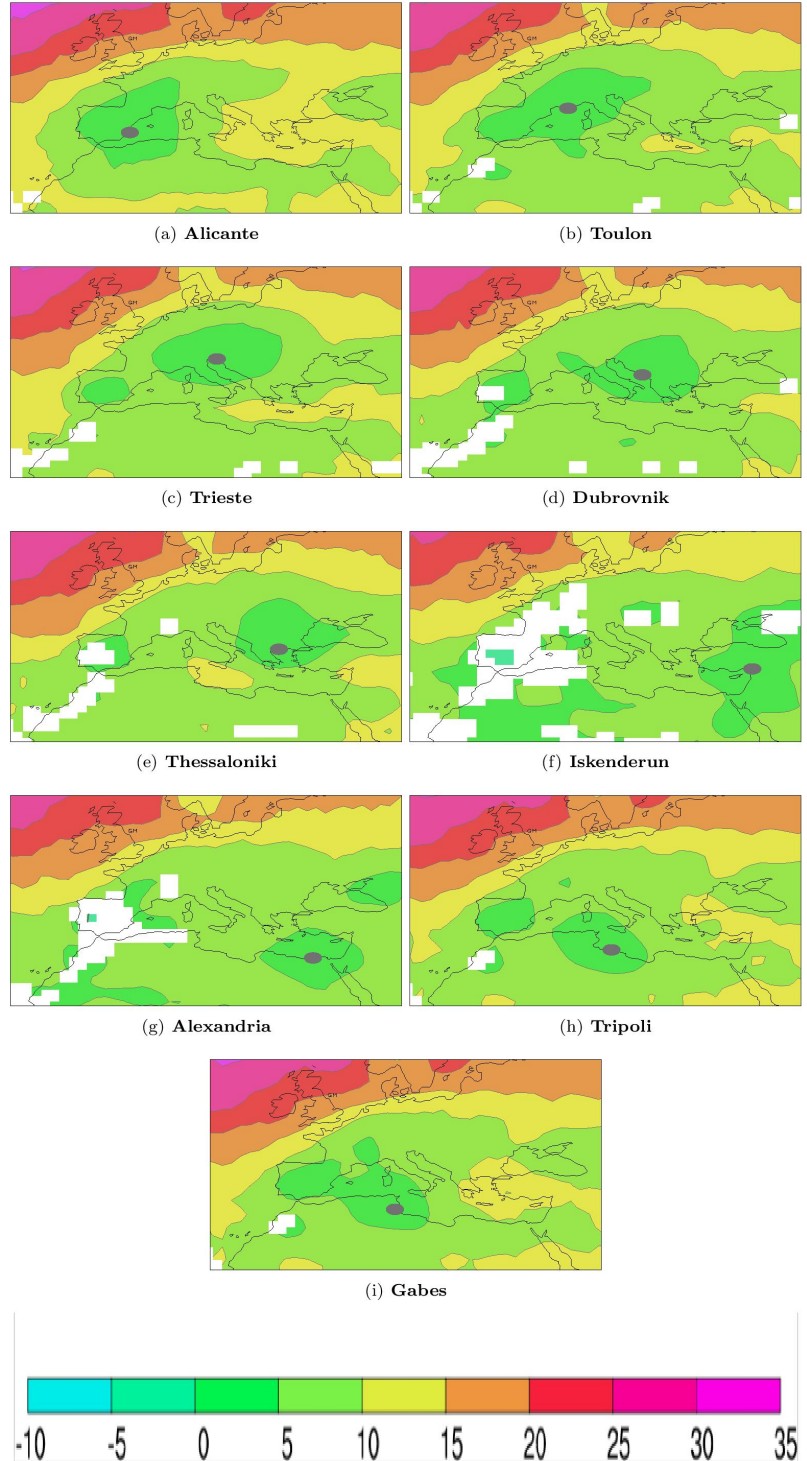

**Figure 10.** Difference between MSLP (hPa) at the coastal station and cyclone pressure minimum as function of cyclone position. Each panel considers a different coastal station: Alicante (a), Toulon (b), Trieste (c), Dubrovnik (d), Thessaloniki (e), Iskenderun (f), Alexandria (g), Tripoli (h), Gabes (h). Only values reaching the 90% significance level are shown.

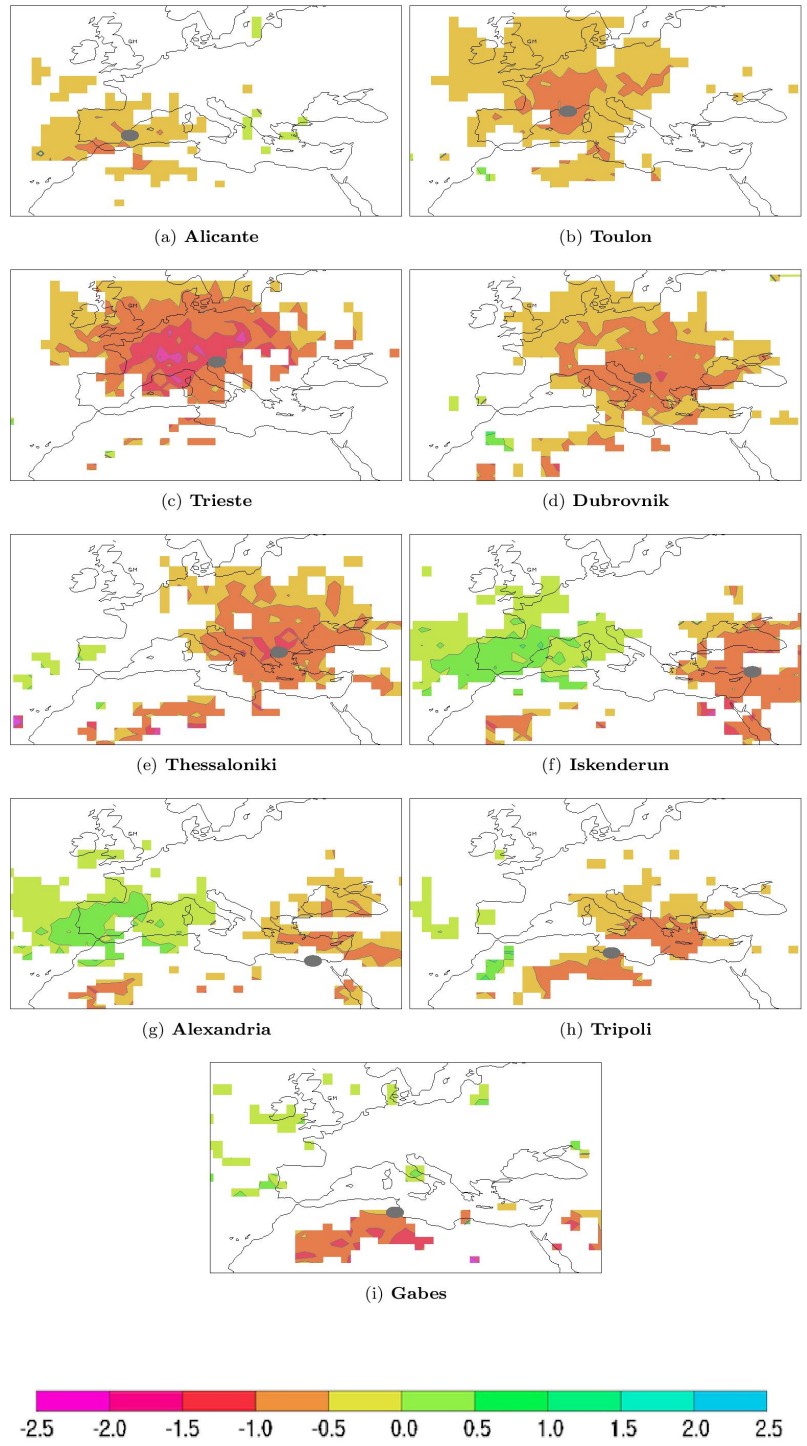

**Figure 11.** Linear regression coefficient (cm hPa−1) of SLAs at the coastal stations versus MSLP cyclone minima as function of the cyclone center position. Each panel considers a different coastal station: Alicante (a) , Toulon (b) , Trieste (c) , Dubrovnik (d) , Thessaloniki (e) , Iskenderun (f) , Alexandria (g), Tripoli (h) , Gabes (h). Only values significant at the 90% confidence level are shown.

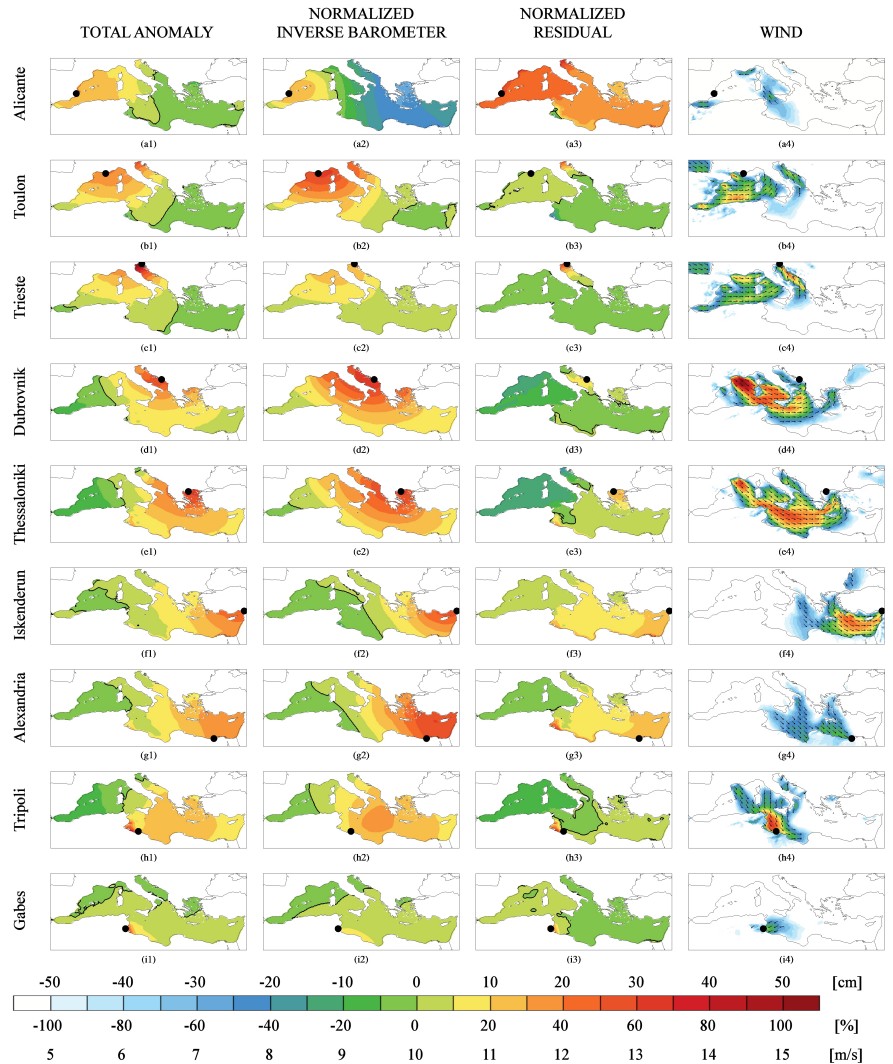

**Figure 12.** Composites of sea level anomalies and wind fields at the time of positive SLAs at the 9 stations considered in this study: Alicante, Toulon, Trieste, Dubrovnik, Thessaloniki, Iskenderun, Alexandria, Tripoli, Gabes (from top to bottom in this order). The first column reports the total anomaly (cm, upper annotation along the color bar), the second column the contribution due to the inverse barometer effect, the third column the residual. Values in the second and third column are normalized with the maxima of the total SLA in the first column (%, first line annotation below the color bar). The thick black line denotes the zero level contour. The fourth column shows the wind field (m/s, lowest annotation below the color bar, values below 5m/s are masked). Arrows that show the wind direction are drawn every $1.25^0$ and only for values larger than 7.5m/s.

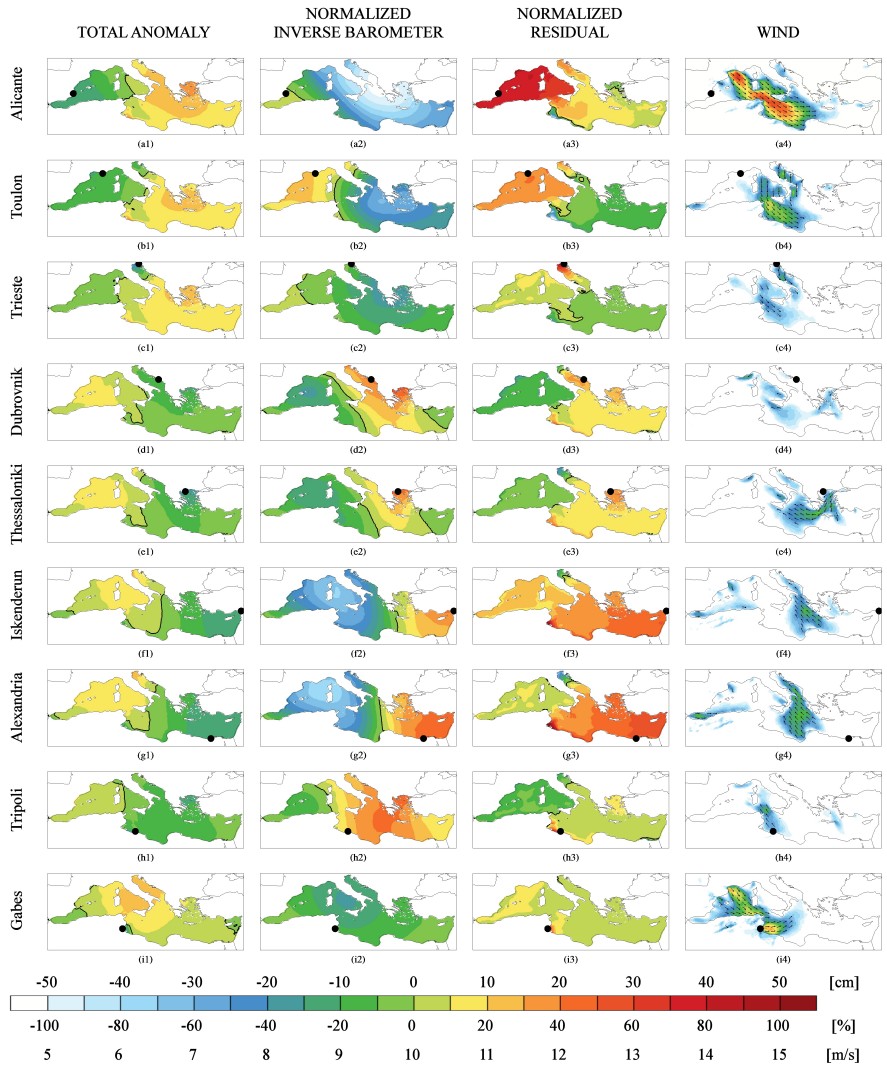

**Figure 13.** same as figure 12 except negative sea level anomaly events are considered (in this case the minima of the SLA total in the first column are used for producing normalized values in the second and third column).