# Peer review of "The effect of cyclones crossing the Mediterranean region on sea level anomalies at the Mediterranean Sea coast"

_Natural Hazards and Earth System Sciences, 2019_

## Referee Comment (RC1) · Anonymous Referee #1 · 22 Feb 2019

This paper (#nhess-2019-6), submitted in EGU's journal *Natural Hazards and Earth System Sciences (NHESS)*, reports on an effort to investigate the effects of Mediterranean cyclones on large sea level anomalies (SLA), both positive and negative, with focus on the coastal zone. SLAs in the Mediterranean are correlated to the intensity and position of cyclonic motions in the atmosphere based on a pre-validated storm track algorithm. Several coastal stations scattered over the basin are used in the analysis, where hindcast SLA modeling results are considered. Main findings refer to the contribution of barometric lows and related strong wind fields especially in shallow water areas with sufficient fetch. The inverse barometer effect is proved to be the prime factor of large positive SLAs at the coast near cyclone centers, whereas negative SLAs at the opposite far away sides of the Mediterranean basin are also attributed to a cross-basin SLP gradient due to the presence of cyclones. Furthermore, negative SLAs are also thought by the authors to be partly caused by residual water mass redistribution within the Mediterranean. Positive SLAs are related to the cyclones' positions and intensities. Overall, cyclogenesis is westernmost in the Mediterranean basin producing positive SLAs enhanced also by Atlantic cyclones. Eastern stations are mostly influenced by the classic Eastern cyclogenesis centers, and North African cyclones mainly induce positive SLAs at the central African coast and negative SLAs at the eastern Mediterranean and North Aegean coasts.

The features of the manuscript could be summed up to the following:

| *ISSUES* | *GRADING* |
| --- | --- |
| Originality / Impact of the Work | Good |
| Amount / Significance of the Work or Results | Good (needs corrections) |
| Acknowledgment of the Work of Others in References | Good (needs additions) |
| Completeness of the Reported Work | Good (needs extra clarifications and further removal or addition of some results) |
| Clarity in Writing | Very Good |
| Clarity in Tables and Graphs | Good |

This is an interesting paper that is well structured and should deserve publication, but only after a major revision. Changes could be included regarding possible additions of new graphs about negative SLAs correlation to anticyclones and barometric highs over the Mediterranean or eventual removal of the try to link negative SLAs in specific locations of the basin to cyclonic motions over other remote parts of it. Moreover some extra clarifications of the approaches followed could enhance the paper's current scientific value. If the authors decide to defend their choice to link negative SLAs to storm tracks rather than to anticyclonic high SLPs in the area of study, then some extra clarifications are needed and/or more convincing explanations should be provided about the soundness of the approach. Moreover set-up/down results could be strengthened by further explanations behind the hydrodynamic response of SLAs to wind patterns in the area. The use of English language is good for a publishable article in *NHESS* (please see comments on a few expressions in Specific Comments).

In the following, I present my basic concerns and some specific comments/questions together with a few editorial changes needed in order to strengthen the manuscript's quality. Major revisions would be required.

**General Comments:**

1) Page 2 Line 20: In the Introduction the authors provide a brief review of storm surges in the Mediterranean and state that "there is little literature considering the synoptic conditions leading to storm surges at other locations and no study has considered negative SLAs", yet there is crucial literature left out from their state-of-the-art. The following references should be added and their basic findings concisely discussed in connection to the present paper's goals:

Bengtsson, L., Hodges, K.I., Roeckner, E. (2006). Storm tracks and climate change. *J. Clim.* 9(15): 3518–3543.

Calafat, F.M., Jordà, G., Marcos, M., Gomis, D. (2012). Comparison of Mediterranean sea level variability as given by three baroclinic models. *J. Geophys. Res.* 117, C02009.

Campins, J., Genovés, A., Picornell, M.A., Jansà, A. (2011). Climatology of Mediterranean cyclones using the ERA-40 dataset. *Int. J. Climatol.* 31(11): 1596–1614.

Makris, C., Galiatsatou, P., Tolika, K., Anagnostopoulou, C., Kombiadou, K., Prinos, P., Velikou, K., Kapelonis, Z., Tragou, E., Androulidakis, Y., Athanassoulis, G., Vagenas, C., Tegoulias, I., Baltikas, V., Krestenitis, Y., Gerostathis, T., Belibassakis, K. and Rusu, E. (2016). Climate Change Effects on the Marine Characteristics of the Aegean and the Ionian Seas. *Ocean Dynamics*, 66(12): 1603–1635.

Marcos, M., Jordà, G., Gomis, D., Pérez, B. (2011). Changes in storm surges in southern Europe from a regional model under climate change scenarios. *Glob. Planet. Change*, 77(3): 116–128.

Vousdoukas, M.I., Voukouvalas, E., Annunziato, A., Giardino, A., Feyen, L. (2016). Projections of extreme storm surge levels along Europe. *Clim. Dyn.*, 47: 3171–3190.

Vousdoukas, M.I., Mentaschi, L., Voukouvalas, E., Verlaan, M., Feyen, L. (2017). Extreme sea levels on the rise along Europe's coasts. *Earths Future*, 5: 304–323.

Fernández-Montblanc, T., Vousdoukas, M.I., Ciavola, P., Voukouvalas, E., Mentaschi, L., Breyiannis, G., Feyen, L. and Salamon, P., 2019. Towards robust pan-European storm surge forecasting. *Ocean Modelling*, 133: 129-144.

2) Page 4 Lines 7-17: The cyclone identification methodology for the Mediterranean is pre-validated, but it is not clear if the specific approach can avoid misrepresentation of storm tracks due to secondary lows, *i.e.* setting an acute angle <85° between two segments of the track defined by three successive points of predicted low pressure centers, in order to consider separate storms. Please see e.g. NASA's storm tracking algorithm (https://data.giss.nasa.gov/stormtracks/). This criterion is usually invoked as the enfeebled extratropical cyclones of the Mediterranean are not found to "double back" on themselves over the course of 6- to 12-hourly timespans. By setting such a limit the possibility of an algorithm to misidentify secondary lows (which can form in the wake of extratropical cyclones) as a reversal of the primary low pressure centers can be avoided. Please further discuss the use of storm identification techniques.

3) Throughout the entire paper, the authors claim that negative SLAs (extensive set-down of coastal sea levels) are attributed to cyclonic motions in the atmosphere in sites practically very far away in the opposite side of the basin, rather than the high pressure barometric systems (anticyclones) during "good weather" over the specific study areas. There is an idea presented that the big negative SLAs are associated with cross-basin SLP gradients, but this seems like a speculation as it is not fully proved and further methods and graphs are need to support the authors' assertions. In the specific comments some recommendations are provided.

4) Moreover no Aeolian regime and wind patterns/vector-maps are given in the study area to uphold the authors' conclusions about negative SLAs induced by wind set-down. Therefore wind roses or other related info should be provided to confirm interesting results of set-up and set-down.

5) Results about positive SLAs are finely reproduced and very interesting, but the correlation of negative SLAs to cyclonic atmospheric motions seems specious. Specifically, there may exist different cyclones (or barometric lows in general) outside of the Mediterranean window presented in the paper (thus not shown in maps), which may develop in regions even closer to the specific study locations compared to classic cyclogenesis centers of the basin, especially in the eastern and southern parts of it. Moreover certainly there exist essential periods of negative SLAs throughout the Mediterranean during good/mild weather with high pressure systems over the entire basin that cannot be linked to a cyclone. These cases refer to mild recession or still water levels of the sea surface in most parts of the

basin, but are overlooked by the authors in their quest to associate extreme barometric lows to negative SLAs.

**Specific Comments:**

*Data and Methods*

Page 3, Line 4: There are surely other important sites on the Mediterranean coastal zone with estimated larger values of SLAs. They should be at least mentioned. Please advise on References provide in General Comment #1.

*Results*

Page 4 Line 30 – Page 5 Line 7 and Figs. 2-3: High correlation coefficients between modelled and observed MSLP composites are well-expected, since ERA-Interim re-analysis data are corrected based on the same in situ observations that the authors use for comparisons. In any way, are the input (atmospheric) data further properly validated or are they evaluate in previous studies? Please elaborate. It would be preferable if the authors used comparisons of modelled SLP fields vs. measurements by meteorological stations unassimilated in the modelled ERA data.

Page 4 Lines 6–15 and Fig. 3: Logically the "large" values of negative SLAs are mostly associated to the huge SLP values (e.g. scoring up to 1025 hPa) over the area, rather than small SLP values in regions far away from the coastal study locations, in opposite sides of the Mediterranean basin.

Page 6 Lines 3–7 and Fig. 5: This comment seems to be based on a misconception of the inverse barometer effect. In what sense is that an exception? Large negative SLAs are consistent with the very large values of MSLP (huge barometric highs of 1025hPa) in all graphs and over vast areas around all study locations.

Page 6 Line 11: Yet again the passage of a deep MSLP minimum over central Europe is likely not the first and basic reason for negative SLAs in North-African coasts, but the anticyclone (high barometric) in atmospheric circulation over these areas.

Page 6 Lines 21-22: This seems like a circumstantial observation and should be backed by wind roses or maps in the area to support the existence of offshore winds over shallow continental shelf.

Page 6 Lines 18-19: With an average velocity of translation of the cyclone center close to 32km/hr (Lionello et al., 2016), for a timespan of 44hr (as top in Fig. 5), you have a movement of the low barometric center of about 1536km, which is still very small compared to the distance in Fig. 5 map.

Page 7 Lines 15-23: This analysis could only be corroborated by correlation to the wind characteristics by PCA method, SOMMS approach and/or other methods of weather pattern identification. It could be omitted if not supported by further analytical comments and results on correlation of SLAs to atmospheric forcing.

Page 7 Lines 29-31: This sentence needs rephrasing. From "showing that he main…" and on this expression is not an explanation or a conclusive remark but a repetition of the first half sentence.

Page 10 Line 31 and Fig. 9: This exactly proves that the inverse barometer effect is mainly responsible for negative SLAs as MSLPs are pretty high over the certain study locations.

Page 11 Line 36 and Fig. 10: 700km are not rendered as large distances in terms of synoptic scale phenomena. Moreover the fact that big distances of the cyclone center donot allow for any influence of the cyclonic low MSLPs to the point-modelled SLAs is proven by Fig. 10, see e.g. green cells in Iskenderun and Alexandria maps. If there was a similar Figure for negative SLAs this would be further strengthened. Or else if such a figure disproves the authors' claims then this kind of analysis should be discarded form the paper.

 Page 12 Lines 23-25 and Fig. 12: This is probably the case, but further wind data in the surrounding area in the Libyan and Adriatic Seas are needed to be shown in order to prove that.

**Technical/Editorial Comments:**

Page 2, Line 20: correct to "Apart from…"

Page 6 Line 20: change "high pressure over most of the basin" to "high pressure over a large part of the basin"

Tables 2-3: Correct to "Dubrovnik"

Page 13 Lines 8, 9 and 33: Correct to "An SLA hindcast…", "along the entire Mediterranean…" and "which is an.." respectively. Page 14 Line 6: Delete one "also".

---

## Referee Comment (RC2) · Anonymous Referee #2 · 27 Feb 2019

In this paper, the authors tried to relate the sea level anomalies at 9 stations in the Mediterranean on a climatological basis with cyclonic tracks, cyclone position and intensity and to further analyse this relationship. The paper is well structured. However, I have to admit that I tried very hard to follow all the methodological steps and to understand some explanations. At some points, verification is required. Furthermore, I have many queries concerning the relationship of negative SLA with cyclones. More specifically: 1. Abstract, page 1, line 2: ". ….. with dynamics involving different factors". I think that this not valid since the authors discuss only the inverse barometer effect. The effect of the wind is also speculated as I will mention in a subsequent comment 2. Section 1, page 2, lines 24-29: the objectives are not clear and robust. In the whole

paragraph, the same objective is actually repeated with other words. 3. Section 2: the hindcast is based on a 2D barotropic model. I think that this allows many simplifications in the results since the temperature variations are not considered. This is an important limitation and could account for the big differences of SLAs in the observed and simulated time series. 4. Page 4, line 15: What the author mean "depth of the cyclone"? 5. Section 3.1, page 4, lines 24: I am really surprised about these results. The differences are enormous!!!!!The authors should comment on that. I am wondering about the reasoning for the hindcast. 6. Sections 3.3-3.4: The association of the SLAs with the density of cyclones is rather arbitrary. For instance: why a radius of 20 degs from the coast station is selected for search of MSLP? Why the computation of the relative frequency is based on 10 deg radius? Why a time step of 10 days is selected? Why the reference point is located in the Ioanian sea based on a subjective criterion? These thresholds should be verified. 7. I am wondering why the negative SPAs are related with cyclones and not with anticyclones. This seems a more realistic thought and approach. 8. I am not convinced about the reliability of the results in sections 3.3 and 3.4. Many findings are speculated and not verified. The positive SLAs could be related with frontal systems that are not considered in this study. 9. Section 3.5, page 11, line 4: why a linear regression is used? A lag correlation should be attempted since the effect of cyclones on the storm surges is not always instant. 10. Section 3.6: The term "dynamics" is not relevant since there is no discussion on the flow regime.

---

## Referee Comment (RC3) · Anonymous Referee #3 · 27 Feb 2019

The manuscript presents an interesting analysis of the relationship between large sea-level anomalies (SLA) and cyclones in the Mediterranean Sea. The assessment is based on the 100 largest positive and negative SLA from a 22y hindcast at 9 coastal sites along the Mediterranean coast; and the characteristics of cyclones that occurred in the Mediterranean region during the hindcast period. The links between the largest SLA and the cyclones intensities and the location of their centers are statistically analyzed in order to associate the SLA to the synoptic atmospheric conditions that generate them. For most of the stations, the results show that large positive SLA are caused by the presence of a cyclone in the area of the station i.e. the large positive SLA are caused by the inverse barometer effect. On the other hand, some large negative SLA

arise from the presence of a cyclone at the other side of the Mediterranean basin, which generates a MSLP gradient along the basin.

The manuscript is well written and structured; the topic is relevant and the results and findings are interesting and relevant. However, I think that there are few important points that should be addressed and I also have few minor comments.

My major comments are:

1) My major concern about the study is related to the variability of the events sample. First, it is not clear to me why 100 events were selected; is this a subjective decision? From the results shown in Table 2 and 3 it is clear that the different synoptic MSLP fields (i.e. both the cyclones associated to different regions and those not assigned) caused both large positive and negative SLA in most of the stations. However, all events are analysed as a single sample and in some cases the SLA or MSLP fields associated to all events are averaged. This can "hide" or weaken the links between the SLA and the MSLP fields due to the variability of the sample. In my opinion, the results would benefit from clustering the events sample in order to reduce the variability, e.g. by a principal component analysis of the MSLP fields associated to the SLA, and performing a separate analysis to each cluster. At least, this issue should be addressed in the discussion.

2) Following the previous comment, it is not clear in some cases how the presented composites are generated. For example, are composites of Figures 4 & 5 showing average values of MSLP fields from all events? Composites shown in Figures 11 and 12 show the total anomaly, is this the sum of all SLAs? Giving the range of the magnitude of the selected SLAs could also give an idea of the variability within the events.

3) I would suggest to add more details about the cyclone characteristics because it is unclear what is the largest MSLP that can be associated to a cyclone center, or what the differences between shallow and depth cyclones are.

4) Regarding negative SLAs, the analysis is only focused on their correlation with the presence of a cyclone in the opposite region of the Mediterranean basin, but I would imagine that large negative SLA will be highly correlated to high pressure systems (which can be supported by the large number of events not assigned to cyclones reported in Table 3).

5) From the analyses presented it is very difficult (or even impossible) to observe the effects of the wind set-up discussed by the authors. For instance, the wind effects can be represented by adding MSLP gradients maps.

Some minor comments:

P2-L26. It is not clear what "their variations" refers to.

P3-L25. Remove one 0 from "20012"

P3-L26. Parenthesis missing after Marcos et al. (2009)")"

P3-L35. Is there any reason for selecting a time window of 120h? e.g. is this value based on any previous study on the duration of storm surges in the region?

Section 3.1. In this section, I am missing the bias between SLAs from the modelled and observed datasets in order to give an idea of the model performance (also e.g. RMSE)

P12-L21. Repetition Tripoli

P12-L29. Change is-> it

P13-L8. Remove n from An

P14-L4. Repetition "also"

Figure 4. The station locations of Toulon and Trieste in the graphs of the first column are shifted.

[Figure]

---

## Author Comment (AC1) · 20 Mar 2019

**Answers to reviewers 2**

**The reviewer writes "I have to admit that I tried very hard to follow all the methodological steps and to understand some explanations. I have many queries concerning the relationship of negative SLA with cyclones"**

We think that the meaning of some parts of the manuscript have been misunderstood by the reviewer. We add here below further explanation to better clarify the meaning of the results (possibly, the text was not sufficiently clear) and the teleconnection linking a negative sea level anomaly to the presence of a cyclone in the opposite part of the basin. Further, we point to some information that is already present in the manuscript, possibly not sufficiently emphasized.

We clarify that we are not denying that high pressure leads to a negative sea level. On this respect, our study confirms the importance of the inverse barometer effect and show that high pressure causes negative SLAs. We propose to add a new paragraph to the conclusions to clarify this:

"*We clarify that we are not denying that high pressure leads to a negative sea level. Our study clearly supports the importance of the local action of the inverse barometer effect for both positive and negative SLAs. The link between large negative SLAs and cyclones that is shown in this study does not describe a local effect, but a teleconnection, supported by a statistical analysis and explained by the large scale structure of the SLP fields. The connection between cyclone in the opposite part of the basin and negative SLAS at the station is mediated by the cross basin pressure gradient and the presence of a high that locally acts according to the inverse barometer effect*".

Further, "*The link between large negative SLAs and cyclones is a statistical concept. It is not meant that cyclones are the cause of large negative anomalies, but that the synoptic condition leading to negative anomalies is frequently associated to the present of a cyclone in the opposite part of the basin.*" This clarification will be added in the conclusions, page 13, line 25.

Moreover, the role of anticyclones is confirmed adding a new text after lines 5-8 of the abstract:
"*The inverse barometer effect produces a positive anomaly at the coast near the cyclone pressure minimum and a negative anomaly at the opposite side of the Mediterranean Sea, because a cross-basin mean sea level pressure gradient is associated to the presence of a cyclone. This often coincides with the presence of an anticyclone above the station, which causes local negative inverse barometer effect*"

The reviewer support his negative evaluation with 10 short points. Here are our rebuttal/clarifications in replies the 10 items present in the review.

**1. Abstract, page 1, line 2: the reviewer writes that the sentence ".. with dynamics involving different factors" is not valid since the authors discuss only the inverse barometer effect.**
**The reviewer thinks that the effect of the wind is also speculated…**

The hindcasts are performed with a dynamical model (HYPSE, Lionello 2005) based on solving the shallow water equations for depth average currents. The model computes the evolution for sea level resulting from the action of surface pressure, wind stress and bottom friction. Therefore, all relevant factors are included in the dynamics leading to the computed SLAs. The inverse barometer effect has the advantage that it can be immediately diagnosed from the SLP field. The residual is due to the wind and eventually non-stationary, effects. The new versions of figures 11 and 12 contains the composite of the wind fields and show how the intensity of the wind blowing onshore and offshore is associated with the large residuals over shallow water areas. We hope that this clarifies the meaning of the sentence and the role of the wind.

**2. The reviewer (referring to Section 1, page 2, lines 24-29) writes that "the objectives are not clear and robust. In the whole paragraph, the same objective is actually repeated with other words."**

The text quoted by reviewer 2 is repeated here below. Bullets points are used to split the sentences and (in red) the subsections to which the text refers are added. We do not see the reason for the strong criticisms of the reviewer. The text in blue might be consider repeating the first sentence and it could eventually be deleted. However, this looks to us to be a question of taste and not a major criticism to the manuscript.

*This study investigates the link of both positive and negative large SLAs along the Mediterranean coastline to the passage of cyclones over the region (figure 1) and describes how SLAs evolve and respond to the presence of cyclones. It includes an analysis:*

- *of the dynamics of SLAs, (section 3.6)*
- *of the synoptic patterns associated with them (section 3.2) and*
- *of their variations with the position where the SLA occurs (section 3.3)*

*It aims at contributing arguments for understanding the link between the variability and evolution of the MR storm track and of SLAs. It*

- *describes position and track of cyclones that are associated with extreme SLA*

*and (section 3.3)*

- *shows the link between their intensity and the magnitude of the corresponding SLAs (section 3.5)*

**3. Section 2: the reviewer thinks that, since the hindcast is based on a 2D barotropic model, this allows many simplifications in the results since the temperature variations are not considered. The reviewer thinks that this is an important limitation and could account for the big differences of SLAs in the observed and simulated time series.**

At page 3, line 32-34, in the manuscript we write that
*a) both observations and model results have been preprocessed using a HPF (High-Pass Filter) with a cutoff frequency of 1/30 days (Conte and Lionello (2013)) in order to cancel long term components (due to change of mass in the MR and steric effects) and to isolate the component that is caused by the short term meteorological forcing.*
Consequently the comment of the reviewer is, in our opinion, already addressed in the text. The adopted procedure is meant to ensure that our results are not substantially influenced by steric effects.

Further, the discrepancies between observed and simulated SLAs have mainly strong implications for their ranking. At page 4, lines23-28 we write that
*In general, the ranking of SLAs in the observed and simulated time series differs substantially. Consequently, the list of the 100 largest observed ("OBS") and of the 100 largest simulated ("MOD") events share only a fraction of events (table 1). The small number of common events is explained by the grouping of the largest SLAs in a relatively narrow range of values, so that small differences in their magnitude may correspond to large differences in their rank. Therefore, inaccuracies of the HYPSE model and of the driving meteorological fields imply substantial differences in ranking between observed and simulated SLAs.*

**4. Page 4, line 15: The reviewer asks what mean "depth of the cyclone" means**
*"The depth of a cyclone is an estimate of the differences between the pressure minimum and the surrounding background value (see Reale and Lionello, 2013 for details on its computation)."* This clarification will be added to the text

**5. Section 3.1, page 4, lines 24: The reviewer is surprised about these results and claims**
**That the differences are enormous! He asks us to comment on that and wonders about the reasoning for the hindcast**.

The explanation of the difference is the text above (see our answers to comment 3). The motivation of the hindcast is to provide long time series of sea level at locations where surges are relevant, but long observed time series are not available

**6. Sections 3.3-3.4: The reviewer writes that "The association of the SLAs with the density of cyclones is rather arbitrary" and asks a series of questions.**

**why a radius of 20 degs from the coast station is selected for search of MSLP?**
The search radius is a subjective choice, resulting from empirical tests.
Anyway, for positive SLAs, the outcomes of the search depends very weakly on this parameter. In fact, the resulting cyclone centers are closely grouped around the station at a distance much smaller than 20degrees (see figure 6). For negative SLAs a small search radius would miss to detect the presence of a cyclone in the basin and would not allow the analysis of the teleconnection between cyclones and negative SLAs.

**Why the computation of the relative frequency is based on 10 deg radius?**
The point for searching a cyclone center and computing the frequencies in table 2 and 3 is not the station, but the reference point around which the cyclone centers concentrate. Therefore, the former search radius and this parameter follow different logics. Ten degrees are about 850 km in the zonal direction in the central areas of the Mediterranean Sea. The average size of cyclones in the Mediterranean is about 500km, of course with non-negligible geographical differences (see table 7 of Lionello et al. 2006 after Trigo et al, 1999) and they move at the average speed of about 180km in six hours (time step of the ERA-Interim data). Consequently, the 10degrees radius is meant to detect cyclones passing close to the reference point at the time of the SLAs.

Trigo, I. F., Davies, T. D., & Bigg, G. R. (1999). Objective climatology of cyclones in the Mediterranean region. J. Climate, 12, 1685–1696.
Lionello P., Bhend J., Buzzi A., Della-Marta P.M., Krichak S., Jansà A., Maheras P., Sanna A., Trigo I.F., Trigo R. (2006). Cyclones in the Mediterranean region: climatology and effects on the environment. In P.Lionello, P.Malanotte-Rizzoli, R.Boscolo (eds) Mediterranean Climate Variability. Amsterdam: Elsevier (NETHERLANDS), 325-372

**Why a time step of 10 days is selected?**
This step is used for extracting independent SLP fields from the hindcast in order to estimate the probability that a cyclone is present. In general, the requirement that samples are independent is essential for a correct estimate to avoid double counting the same cyclone several times. Lionello et al 2016 show that in the Mediterranean cyclones lasting more than 5 days are extremely rare. Therefore, this step ensures that the climatological probability is estimated using independent samples and every cyclone is counted only once.

**Why the reference point is located in the Ionian sea based on a subjective criterion?**
First, it is important to note that the reference point has been located subjectively only in Iskenderun for negative SLAs. In all other cases "The reference position is the center of the 5deg wide lat-lon cell where the density of cyclone centers has a maximum." (lines 6-8 at page 8). For negative SLAs at Iskenderun (In this case cyclone centers are rather sparsely distributed) this criterion would locate the reference center at the eastern boundary of the map (lines 8-9), downstream of the Mediterranean region. To avoid this, the secondary maximum (largest value after the actual maximum) has been used leading to the point located in the Ionian basin.

**7. The reviewers asks "why the negative SLAs are related with cyclones and not with anticyclones. This seems a more realistic thought and approach**. "

A new figure and a table have been produced to describe the role of anticyclones
"*Figure new1 shows the centers of anticyclones at the time of negative SLAs. It is made following the same procedure that has been used for figure 6, which refers to cyclones. It reveals the location of centers of anticyclones in the areas where figure 5 shows high pressure systems. Anticyclones are actually concentrated around the stations, with the exception of Gabes and, to a lesser degree, Trieste, where the wind effect is much larger than the inverse barometer effect and anticyclones play a minor role. Therefore, negative SLAs are linked to the presence of a high pressure around the station. This is necessarily true for most stations, because of the inverse barometer effect. However (see table new1), the probability to find an anticyclone at a distance lower than 10 degrees from the reference position\* at the time of negative SLAs is significantly larger than the climatological value only for three stations (Toulon, Thessaloniki, Iskenderun). On the contrary, in Gabes, the absence of an anticyclone is linked to negative SLAs (and this is justified by the dominant role of the wind at this station). The link with the presence of a cyclone in the part of the basin opposite to the station (table*

*3) is stronger than what shown in table new1 for anticyclones."* This explanation will be added to the manuscript at the end of section 3.4.

Further, an objective of this study is to show that a robust teleconnection, which is supported by a statistical analysis, links negative SLAs at some stations to the presence of a cyclone in the opposite part f the basin. This link does not describe a local effect. In fact, the connection between cyclone in the opposite part of the basin and negative SLAS at the station is mediated by the cross-basin pressure gradient and the presence of a high that locally acts according to the inverse barometer effect. We anticipated at the beginning of this rebuttal that new text to clarify this will be added to the manuscript

*: The reference position is defined as the center of the 5deg wide lat-lon cell where the density of anticyclone centres (blue square in figure new1) has a maximum (same procedure that was adopted for table 2).*

**8. The reviewer is "not convinced about the reliability of the results in sections 3.3 and 3.4". According to him "Many findings are speculated and not verified. The positive SLAs could be related with frontal systems that are not considered in this study.**

The argument why the reviewer is not convinced are not given. It is therefore a bit difficult to argue against her/his reluctance to accept the content of these sections or propose changes/additions to convince him. He does not say which "findings are speculated and not verified". The results in table 2 and 3 are checked for statistical significance at the 95% confidence level. Mid latitude cyclones are characterized by the present of cold, warm, and occluded fronts, therefore, when considering a cyclone, the action of the fronts associated with it are included.

**9. Section 3.5, page 11, line 4: The reviewer asks "why a linear regression is used? A lag correlation should be attempted since the effect of cyclones on the storm surges is not always instant."**

A linear regression is the simplest tool to model the relationship between a dependent variable (in this case the SLAs) and an explanatory variable (in this case the SLP minimum). The two variables are sometime called predictand and predictor respectively. In this case a statistically significant linear relation is found and it shows hot the SLA levels are linked to the intensity (Minimum SLP) of the cyclones. We agree that this approach ignore the possibility of delayed effects, and therefore may conceptually underestimate the strength of the link (that could be stronger, not weaker, than what we have found). Therefore the approach is successful and we have no reason for rejecting it. Certainly, other approaches to reach the same goal are possible. The correlation intrinsically refers to two time series to be correlated. Here, there are no time series, but a set of SLP minima (predictors) and of SLAs (predictand).

**10. Section 3.6: The term "dynamics" is not relevant since there is no discussion on the flow regime.**
Dynamics is a branch of physical science and subdivision of mechanics that is concerned with the motion of material objects in relation to the physical factors that affect them (Encyclopedia Britannica). In section 3.6 we describe the resulting position of the sea surface (SLA) in relation to the sea level pressure and the wind fields that cause its motion. In our view the term dynamics is justified.

---

## Author Comment (AC2) · 20 Mar 2019

**General answer to reviewer 3.**

We thank Reviewer 3 for her/his comments on our manuscript. Here below are our replies to her/his major concerns (see items 1 to 5 below) and the description of the added material that we have produced for properly addressing them. Some of these concerns/suggestions are shared by reviewer 1, particularly on the need of more explanations describing a) the role of the wind and b) the link to anticyclones. In fact, we present here some new material to address these two issues, which is also present in our answers to reviewer 1:

**1) The reviewer is concerned on the motivation for the selection of 100 samples, their**

**representativeness and the possibility of splitting the overall sampling in different subsets.**

We admit that "The selection of 100 hundred events is a subjective decision. Considering that hindcast covers 22 years, this corresponds to an approximate average of 5 events per year. In the case of Venice this is close to the 80th percentile of the surge events (Lionello et al, 2012\*). Empirical tests have shown that results do not appreciably change using a smaller sample."

This sentence has to be added to the data and methods section.

Splitting the samples in subsets using statistical techniques such as PCA or clustering is certainly a possibility. However, in our study "the internal variability of the sample is explored considering the analysis of the cyclogenesis, which allow to distinguish the different evolutions of cyclones. We have adopted this processoriented approach. In our opinion, it is very plausible that PCA or clustering of the trajectories would have produced very similar outcomes. We admit that this approach might hide some aspects of the internal variability of the sample related to different synoptic patterns at the time of the SLAs, which might be worth to explore in a future studies for those stations where this issue would eventually result significant." This paragraph will be added to the "Discussion and conclusion" section

\*Lionello P, Cavaleri L, Nissen KM, Pino C, Raicich F, Ulbrich U (2012) Severe marine storms in the Northern Adriatic: Characteristics and trends. *Phys Chem Earth*, 40-41:93-105, doi:10.1016/j.pce.2010.10.002

**2**) We confirm that the composites of Figures 4 & 5 show (for each station) average values of MSLP fields from all events. Analogously, composites in Figures 11 and 12 show the average value of the anomaly and its components (Inverse barometer effect and residual). A further column has been added with the composites of the wind fields.

**3)** Reviewer suggests to add more details about the cyclone characteristics because it is unclear what is the largest MSLP that can be associated to a cyclone center, or what the differences between shallow and depth cyclones are.**

Indeed, the adopted tracking algorithm provides further information that it is useful to add. We have prepared four tables (Tables SuM1-SuM4, see below) that show for each cyclogenesis area the mean values of the central SLP minimum and of the depth of the cyclone with the respective standard deviations. We mean to add them a supplementary material. The results show that cyclones generated over the Atlantic are deeper and with a lower central SLP minimum than those generated in other areas.

4) The reviewer asks for analyzing the link of negative SLAS with high pressure systems

A new figure (see below figure new1) has been produced that describes the role of anticyclones and a new table (see table new1 below) have been produced. "Figure new1 shows the centers of anticyclones at the time of negative SLAs. It is made following the same procedure that has been used for figure 6, which refers to cyclones. It reveals the location of centers of anticyclones in the areas where figure 5 shows high pressure systems. Anticyclones are actually concentrated around the stations, with the exception of Gabes and, to a lesser degree, Trieste, where the wind effect is much larger than the inverse barometer effect and anticyclones play a minor role. Therefore, negative SLAs are linked to the presence of a high pressure around the station. This is necessarily true for most stations, because of the inverse barometer effect. However (see table new1), the probability to find an anticyclone at a distance lower than 10 degrees from the reference position\* at the time of negative SLAs is significantly larger than the climatological value only for three stations (Toulon, Thessaloniki, Iskenderun). On the contrary, in Gabes, the absence of an anticyclone is linked to negative SLAs (and this is justified by the dominant role of the wind at this station). The link with the presence of a cyclone

in the part of the basin opposite to the station (table 3) is stronger than what shown in table new1 for anticyclones." This explanation will be added to the manuscript at the end of section 3.4.

Further, we will extend the sentence at lines 5-8 of the abstract as it follows: "The inverse barometer effect produces a positive anomaly at the coast near the cyclone pressure minimum and a negative anomaly at the opposite side of the Mediterranean Sea, because a cross-basin mean sea level pressure gradient is associated to the presence of a cyclone. This often coincides with the presence of an anticyclone above the station, which causes local negative inverse barometer effect"

We clarify that we are not denying that high pressure leads to a negative sea level because of the inverse barometer effect and our study clearly supports its importance. The fifth paragraph at the beginning of this answers has to be added to the conclusions to clarify this.

\*: The reference position is defined as the center of the 5deg wide lat-lon cell where the density of anticyclone centers (blue square in figure new1) has a maximum (same procedure that was adopted for table 2).

5) The reviewer asks for adding material to document the effect of the wind. This is indeed a useful suggestion and two columns with the wind composites at the time when SLAs are largest have been added (see figures 11 and 12 below).

"The action of the wind is evident in the fourth column of figures 11 and 12, which show the composites of the wind fields at the time of the largest anomaly. In these maps, the presence of a strong wind blowing towards the coast (fig.11, positive SLAs) or offshore (fig.12, negative SLAs) is consistent with the large residuals at Trieste, Tripoli and Gabes. For positive SLAs is also present in correspondence with the residuals (which smaller than in the previous stations) at Alexandria, Iskenderun and Thessaloniki." These sentences will be added at the end of section 3.6.

| Station      | P SLA+ | P CLIM+ | P SLA+ |
|--------------|-------------------|--------------------|-------------------|
| ALICANTE     | 38                | 40                 | 62                |
| TOULON       | 56                | 40                 | 44                |
| TRIESTE      | 44                | 37                 | 54                |
| DUBROVNIK    | 42                | 37                 | 58                |
| THESSALONIKI | 62                | 41                 | 38                |
| ISKENDERUN   | 51                | 31                 | 49                |
| ALEXANDRIA   | 43                | 39                 | 57                |
| GABES        | 33                | 44                 | 67                |
| TRIPOLI      | 40                | 48                 | 60                |

**Table new1**. Statistics of cyclones producing the 100 largest negative sea level anomalies in each considered station. The two columns labelled " $P_{SLA+}$ " and " $P_{clim-}$ ", report the probability (%) to find an anticyclone within a 10degs search radius from the reference point (denoted with a yellow square in figure new1) at the time of the event and the corresponding climatological mean value, respectively. Bold values denote differences between the "PSLA+" and "Pclim-" that are statistically significant at the 95% level. The last column reports the number of events in the period 1979-2001 that were not assigned to any anticyclone

---

## Author Comment (AC3) · 20 Mar 2019

**General answer to reviewer 1.**

We thank Reviewer 1 for having carefully read our manuscript and for her/his comments. We agree on the suggestions to add material describing a) the role of the wind and b) the link to anticyclones. On this respect, we think that this review really helps improving the clarity and the quality of the manuscript. Some of these concerns/suggestions are shared by reviewer 3. Therefore, there is an overlap between our answers to the two reviewers and the new material that we present here is duplicated in our answer to reviewer 3.

Our feedback to suggestions a) and b) and answers to the request for clarifying some issues are here below. They refer to the supplement, where reviewer 1 has described her/his general concerns and provided suggestions for improving the manuscript.

Note that our study shows that in shallow areas the wind is a main factor leading to sea level anomalies (SLA). This occurs particularly at the stations (Trieste and Gabes) located in the areas where anomalies have the largest values in the Mediterranean Sea. In order to stress this, a new column is added to former figures 11 and 12 showing the composites of the surface wind fields at the time of the maximum SLA. The new column clearly shows the correspondence between wind fields and the residual SLA, which is not produced by the inverse barometer effect.

Moreover, maps describing the position of high pressures and a table documenting the frequency of their presence when large negative SLAs occur are presented. This new information shows that the link with the presence of a high pressure system is indeed present, but weaker than the connection with the presence of cyclones in the opposite part of the Mediterranean Sea for some stations.

Finally, "we clarify that we are not denying that high pressure leads to a negative sea level. Our study clearly supports the importance of the local action of the inverse barometer effect for both positive and negative SLAs. The link between large negative SLAs and cyclones that is shown in this study does not describe a local effect, but a teleconnection, supported by a statistical analysis and explained by the large scale structure of the SLP fields. The connection between cyclone in the opposite part of the basin and negative SLAS at the station is mediated by the cross basin pressure gradient and the presence of a highpressure that locally acts according to the inverse barometer effect". This paragraph will be added to the "Discussion and Conclusion section".

**Answers to general comments 1-5 of reviewer 1**

Here below our answers to the comments of the reviewer. At the same time we describe where and how our manuscript would be changed to clarify the results of our study. Slant characters denote the text to be added to the manuscript

1) The reviewer lists very interesting papers on storm tracks and modelling of surges, several of them describing impacts of climate change on sea level anomalies. Those papers are very interesting, but they do not consider the description of the link between synoptic patterns and SLAs in the Mediterranean Sea, which is the actual object of this study. Mainly they describe the results of simulations that describe evolution and change of storm surges. They do not address explicitly the link between synoptic features and SLAs.

2) We agree that to add to the manuscript some details on the used tracking algorithm is useful, "This cyclone tracking algorithm contains features that are meant to detect the formation of cyclones inside the Mediterranean and, at the same time, to avoid the inflation of the number of cyclones, determined by considering small, short lived feature as independent systems. This is a crucial balance as a large fraction of Mediterranean cyclones are secondary lows triggered by the presence of a large system over north and central Europe The method first partitions the SLP field in depressions, which can be considered candidates for independent cyclones, by merging all steepest descent paths leading to the same minimum. The small depressions that share a boundary with a deeper depression are included in the latter to form a single cyclone. The position of the cyclone is computed as the average of the points with SLP not more than 3 hPa higher than the actual minimum to compensate for large deviation of cyclones from the circular shape. Finally, when searching for successive positions of cyclones to construct their track, the search area is shifted southeasterly with respect to the former center (see Lionello et al., 2002 and Reale and Lionello, 2013, for more details)".

This text will be added to the manuscript. The "double back" of cyclones trajectories (mentioned by the reviewer) is not evident in any of the former applications of this method, probably because of the specific features of this method.

3) "The link between large negative SLAs and cyclones is a statistical concept. It is not meant that cyclones are the cause of large negative anomalies, but that the synoptic condition leading to negative anomalies is frequently associated to the present of a cyclone in the opposite part of the basin." This clarification will be added in the conclusions, page 13, line 25.

The link between the presence of a cyclone and the cross-basin pressure gradient is described in figure 9. This figure (described in the paragraph beginning at line 31 page 10) shows the difference of MSLP between the station and the cyclone center. "*The presence of positive values, between 10 and 15hPa in the areas of the basin opposite to the station evidences that when the cyclone center is located in such areas the pressure at the station is high and the inverse barometer effect contributes to negative sea level anomalies.*" Note that almost all differences in the maps are statistically significant. We admit that this was not clearly described in the text and the sentence between quotation marks will be added the manuscript.

4) We agree to document better the relevance of the wind. Two new columns in figure 11 and 12 are meant to provide the evidence of the action of the wind (see figures below). "The action of the wind is evident in the fourth column of figures 11 and 12, which show the composites of the wind fields at the time when the SLAs are largest anomaly. In these maps, the presence of a strong wind blowing towards the coast (fig.11, positive SLAs) or offshore (fig.12, negative SLAs) is consistent with the large residuals at Trieste, Tripoli and Gabes. For positive SLAs is also present in correspondence with the residuals (which smaller than in the previous stations) at Alexandria, Iskenderun and Thessaloniki." These sentences will be added at the end of section 3.6.

5) A new figure has been produced to answer to this comment of the reviewer and describe the role of anticyclones. "Figure new1 shows the centers of anticyclones at the time of negative SLAs. It is made following the same procedure that has been used for figure 6, which refers to cyclones. It reveals the location of centers of anticyclones in the areas where figure 5 shows high pressure systems. Anticyclones are actually concentrated around the stations, with the exception of Gabes and, to a lesser degree, Trieste, where the wind effect is much larger than the inverse barometer effect and anticyclones play a minor role. Therefore, negative SLAs are linked to the presence of a high pressure around the station. This is necessarily true for most stations, because of the inverse barometer effect. However (see table new1), the probability to find an anticyclone at a distance lower than 10 degrees from the reference position\* at the time of negative SLAs is significantly larger than the climatological value only for three stations (Toulon, Thessaloniki, Iskenderun). On the contrary, in Gabes, the absence of an anticyclone is linked to negative SLAs (and this is justified by the dominant role of the wind at this station). The link with the presence of a cyclone in the part of the basin opposite to the station (table 3) is stronger than what shown in table new1 for anticyclones." This explanation will be added to the manuscript at the end of section 3.4.

Further, we will extend the sentence at lines 5-8 of the abstract as it follows: "The inverse barometer effect produces a positive anomaly at the coast near the cyclone pressure minimum and a negative anomaly at the opposite side of the Mediterranean Sea, because a cross-basin mean sea level pressure gradient is associated to the presence of a cyclone. This often coincides with the presence of an anticyclone above the station, which causes local negative inverse barometer effect"

We clarify that we are not denying that high pressure system lead to a negative sea level. Our study clearly supports the importance of them and of the inverse barometer effect. The fifth paragraph at the beginning of this answers has to be added to the conclusions to clarify this.

\*: The reference position is defined as the center of the 5deg wide lat-lon cell where the density of anticyclone centres (blue square in figure new1) has a maximum (same procedure that was adopted for table 2).

| Station      | Psla+ | Pclim+ | Psla+ |
|--------------|-------|--------|-------|
| ALICANTE     | 38    | 40     | 62    |
| TOULON       | 56    | 40     | 44    |
| TRIESTE      | 44    | 37     | 54    |
| DUBROVNIK    | 42    | 37     | 58    |
| THESSALONIKI | 62    | 41     | 38    |
| ISKENDERUN   | 51    | 31     | 49    |
| ALEXANDRIA   | 43    | 39     | 57    |
| GABES        | 33    | 44     | 67    |
| TRIPOLI      | 40    | 48     | 60    |

**Table new1**. Statistics of cyclones producing the 100 largest negative sea level anomalies in each considered station. The two columns labelled "PSLA+" and "Pclim.", report the probability (%) to find an anticyclone within a 10degs search radius from the reference point (denoted with a yellow square in figure new1) at the time of the event and the corresponding climatological mean value, respectively. Bold values denote differences between the "PSLA+" and "Pclim-" that are statistically significant at the 95% level.

---

## Author Response (AR1)

Dear Editor:

Our answers to the comments of the reviewers are attached. They correspond to the contents of our answers to the individual reviewers that we have uploaded in the online discussion.

The revised version of the manuscript emphasizes in red the new text and obscures with slant bars the deleted text. We have done this for helping you and the reviewers to identify changes. Please let us know whether you consider this an acceptable presentation of the revised manuscript.

Thanks for your editorial work on this manuscript.

Best regards

Piero Lionello of the behalf of all authors

**Answer to reviewer 1.**

**Reviewer: This is an interesting paper that is well structured and should deserve publication, but only after a major revision. Changes could be included regarding possible additions of new graphs about negative SLAs correlation to anticyclones and barometric highs over the Mediterranean or eventual removal of the try to link negative SLAs in specific locations of the basin to cyclonic motions over other remote parts of it. Moreover some extra clarifications of the approaches followed could enhance the paper's current scientific value. If the authors decide to defend their choice to link negative SLAs to storm tracks rather than to anticyclonic high SLPs in the area of study, then some extra clarifications are needed and/or more convincing explanations should be provided about the soundness of the approach. Moreover set-up/down results could be strengthened by further explanations behind the hydrodynamic response of SLAs to wind patterns in the area. The use of English language is good for a publishable article in _NHESS_ (please see comments on a few expressions in Specific Comments). In the following, I present my basic concerns and some specific comments/questions together with a few editorial changes needed in order to strengthen the manuscript's quality. Major revisions would be required.**

We thank Reviewer 1 for having carefully read our manuscript and for her/his comments. We agree on the suggestions to add material describing: a) the role of the wind and b) the link to anticyclones. On this respect, we think that this review really helps improving the clarity and the quality of the manuscript. Some of these concerns/suggestions are shared by reviewer 3. Therefore, there is an overlap between our answers to the two reviewers and some material that we present here is duplicated in our answer to reviewer 3.

Our feedback to suggestions a) and b) and answers to the request of clarifying some issues are here below. They refer to the supplement material (submitted by reviewer 1) with her/his general concerns and suggestions for improving the manuscript.

Our study shows that in shallow areas the wind is a main factor leading to sea level anomalies (SLA). This occurs particularly at the stations (Trieste and Gabes) located in the areas where anomalies have the largest values in the Mediterranean Sea. In order to emphasize this, a new column has been added to former figures 11 and 12 with composites of the surface wind fields at the time of the maximum SLA. The new column clearly shows the correspondence between wind fields and the residual SLA, which is the SLA component not produced by the inverse barometer effect.

Moreover, maps describing the position of anticyclones and a table documenting the frequency of their presence when large negativeSLAs occur have been added to our manuscript. This new information shows the link with the presence of a high pressure system, but generally it is weaker than the connection with the presence of cyclones in the opposite part of the Mediterranean Sea.

Finally, "*we clarify that we are not denying that high pressure leads to a negative sea level. Our study clearly supports the importance of the local action of the inverse barometer effect for both positive and negative SLAs. The link between large negative SLAs and cyclones that is shown in this study does not describe a local effect, but a teleconnection, supported by a statistical analysis and explained by the large scale structure of the SLP fields. The connection between cyclone in the opposite part of the basin and negative SLAS at the station is mediated by the cross basin pressure gradient and the presence of a high pressure that locally acts according to the inverse barometer effect*". This paragraph has been added to the "Discussion and Conclusion section".

**Answers to general comments 1-5 of reviewer 1**

Here below our answers to the comments of the reviewer (bold characters) and the changes that have been implemented to clarify the results of our study. The text that has been added to the manuscript is denoted with slant characters in this reply and is marked with red in the manuscript.

**1) Reviewer: Page 2 Line 20: In the Introduction the authors provide a brief review of storm surges in the Mediterranean and state that "there is little literature considering the synoptic conditions leading to storm surges at other locations and no study has considered negative SLAs", yet there is crucial literature left out from their state-of-the-art. The following references should be added and their basic findings concisely discussed in connection to the present paper's goals:**

**Bengtsson, L., Hodges, K.I., Roeckner, E. (2006). Storm tracks and climate change. J. Clim. 9(15): 3518–3543.**

**Calafat, F.M., Jordà, G., Marcos, M., Gomis, D. (2012). Comparison of Mediterranean sea level variability as given by three baroclinic models. J. Geophys. Res. 117, C02009.**

**Campins, J., Genovés, A., Picornell, M.A., Jansà, A. (2011). Climatology of Mediterranean cyclones using the ERA-40 dataset. Int. J. Climatol. 31(11): 1596–1614.**

**Makris, C., Galiatsatou, P., Tolika, K., Anagnostopoulou, C., Kombiadou, K., Prinos, P., Velikou, K., Kapelonis, Z., Tragou, E., Androulidakis, Y., Athanassoulis, G., Vagenas, C., Tegoulias, I., Baltikas, V., Krestenitis, Y., Gerostathis, T., Belibassakis, K. and Rusu, E. (2016). Climate Change Effects on the Marine Characteristics of the Aegean and the Ionian Seas. Ocean Dynamics, 66(12): 1603–1635.**

**Marcos, M., Jordà, G., Gomis, D., Pérez, B. (2011). Changes in storm surges in southern Europe from a regional model under climate change scenarios. Glob. Planet. Change, 77(3): 116–128.**

**Vousdoukas, M.I., Voukouvalas, E., Annunziato, A., Giardino, A., Feyen, L. (2016). Projections of extreme storm surge levels along Europe. Clim. Dyn., 47: 3171–3190.**

**Vousdoukas, M.I., Mentaschi, L., Voukouvalas, E., Verlaan, M., Feyen, L. (2017). Extreme sea levels on the rise along Europe's coasts. Earths Future, 5: 304–323.**

**Fernández-Montblanc, T., Vousdoukas, M.I., Ciavola, P., Voukouvalas, E., Mentaschi, L., Breyiannis, G., Feyen, L. and Salamon, P., 2019. Towards robust pan-European storm surge forecasting. Ocean Modelling, 133: 129-144.**

The reviewer lists very interesting papers on storm tracks and modelling of surges, several of them describing impacts of climate change on sea level anomalies. Those papers are very interesting, but they do not consider the description of the link between synoptic patterns and SLAs in the Mediterranean Sea, which is the actual object of this study. Mainly they describe the results of simulations that describe evolution and change (depending on scenarios) of storm surges. They do not address explicitly the link between atmospheric synoptic features and SLAs.

**2) Reviewer: Page 4 Lines 7-17: The cyclone identification methodology for the Mediterranean is pre-validated, but it is not clear if the specific approach can avoid misrepresentation of storm tracks due to secondary lows, i.e. setting an acute angle <85° between two segments of the track defined by three successive points of predicted low pressure centers, in order to consider separate storms. Please see e.g. NASA's storm tracking algorithm (https://data.giss.nasa.gov/stormtracks/). This criterion is usually invoked as the enfeebled extratropical cyclones of the Mediterranean are not found to "double back" on themselves over the course of 6- to 12-hourly timespans. By setting such a limit the possibility of an algorithm to misidentify secondary lows (which can form in the wake of extratropical cyclones) as a reversal of the primary low pressure centers can be avoided. Please further discuss the use of storm identification techniques.**

We agree that to add to the manuscript some details on the used tracking algorithm is useful, *"This cyclone tracking algorithm contains features that are meant to detect the formation of cyclones inside the Mediterranean and, at the same time, to avoid the inflation of the number of cyclones, determined by considering small, short lived feature as independent systems. This is a crucial balance as a large fraction of Mediterranean cyclones are secondary lows triggered by the presence of a large system over north and central Europe The method first partitions the SLP field in depressions, which can be considered candidates for independent cyclones, by merging all steepest descent paths leading to the same minimum.The small depressions that share a boundary with a deeper depression are included in the latter to form a single cyclone. The position of the cyclone is computed as the average of the points with SLP not more than 3 hPa higher than the actual minimum to compensate for large deviation of cyclones from the circular shape. Finally, when searching for successive positions of cyclones to construct their track, the search area is shifted southeasterly with respect to the former center (see Lionello et al., 2002 and Reale and Lionello, 2013, for more details)"*.This text has been added to the manuscript at page 4 from line 29. The "double back" of cyclones trajectories (mentioned by the reviewer) is not evident in any of the former applications of this method, probably because of the specific features of this method.

**3) Reviewer: Throughout the entire paper, the authors claim that negative SLAs (extensive set-down of coastal sea levels) are attributed to cyclonic motions in the atmosphere in sites practically very far away**

in the opposite side of the basin, rather than the high pressure barometric systems (anticyclones) during "good weather" over the specific study areas. There is an idea presented that the big negative SLAs are associated with cross-basin SLP gradients, but this seems like a speculation as it is not fully proved and further methods and graphs are need to support the authors' assertions. In the specific comments some recommendations are provided.

*"The link between large negative SLAs and cyclones is a statistical concept. It is not meant that cyclones are the cause of large negative anomalies, but that the synoptic condition leading to negative anomalies is frequently associated to the present of a cyclone in the opposite part of the basin."* This clarification has been addedtothe conclusions, page 16 from line 1.

The link between the presence of a cyclone and the cross-basin pressure gradient is described in figure 10. This figure (described in the paragraph beginning at page 12 line 15) shows the difference of MSLP between the station and the cyclone center as a function of the cyclone position. *"The positive values, between 10 and 15hPa in the areas of the basin opposite to the station evidence that, when the cyclone center is located in such areas, the pressure at the station is high and the inverse barometer effect contributes to negative sea level anomalies."* Note that almost all differences in the maps are statistically significant. We admit that this was not clearly described in the text and the sentence between quotation marks has been added the manuscript, with a consequent minor change in the following sentence.

**4)Reviewer: Moreover no Aeolian regime and wind patterns/vector-maps are given in the study area to uphold the authors' conclusions about negative SLAs induced by wind set-down. Therefore, wind roses or other related info should be provided to confirm interesting results of set-up and set-down.**

We agree to document better the relevance of the wind. Two new columns in figure 12 and 13are meant to provide the evidence of the action of the wind (see figures below). *"The action of the wind is evident in the fourth column of figures 12 and 13, which show the composites of the wind fields at the time when the SLAs are largest anomaly. In these maps, the presence of a strong wind blowing towards the coast (fig.12, positive SLAs) or offshore (fig.13, negative SLAs) is consistent with the large residuals at Trieste, Tripoli and Gabes. For positive SLAs the wind is also present in correspondence with the residuals (which are smaller than in the previous stations) at Alexandria, Iskenderun and Thessaloniki."*These sentences has been addedat the end of section3.6.

**5)Reviewer: Results about positive SLAs are finely reproduced and very interesting, but the correlation of negative SLAs to cyclonic atmospheric motions seems specious. Specifically, there may exist different cyclones (or barometric lows in general) outside of the Mediterranean window presented in the paper (thus not shown in maps), which may develop in regions even closer to the specific study locations compared to classic cyclogenesis centers of the basin, especially in the eastern and southern parts of it. Moreover certainly there exist essential periods of negative SLAs throughout the Mediterranean during good/mild weather with high-pressure systems over the entire basin that cannot be linked to a cyclone. These cases refer to mild recession or still water levels of the sea surface in most parts of the basin, but are overlooked by the authors in their quest to associate extreme barometric lows to negative SLAs.**

A new figure has been produced to answer to this comment of the reviewer and describe the role of anticyclones. *"Figure 8 shows the centers of anticyclones at the time of negative SLAs. It is made following the same procedure that has been used for figure 7, which refers to cyclones. It reveals the location of centers of anticyclones in the areas where figure 5 shows high pressure systems. Anticyclones are actually concentrated around the stations, with the exception of Gabes and, to a lesser degree, Trieste, where the wind effect is much larger than the inverse barometer effect and anticyclones play a minor role. Therefore, negative SLAs are linked to the presence of a high pressure around the station. This is necessarily true for most stations, because of the inverse barometer effect. However (see table 4), the probability to find an anticyclone at a distance lower than 10 degrees from the reference position\* at the time of negative SLAs is significantly larger than the climatological value only for three stations (Toulon, Thessaloniki, Iskenderun). On the contrary, in Gabes, the absence of an anticyclone is linked to negative SLAs (and this is justified by the dominant role of the wind at this station). The link with the presence of a cyclone in the part of the basin opposite to the station (table 3) is stronger than what shown in table new1 for anticyclones."* This explanation has been added to the manuscript at the end of section 3.4.

Further, we have extended the sentence at lines 5-8 of the abstract as it follows: *"The inverse barometer effect produces a positive anomaly at the coast near the cyclone pressure minimum and a negative anomaly at the*

*opposite side of theMediterranean Sea, because a cross-basin mean sea level pressure gradient is associated to the presence of a cyclone. This often coincides with the presence of an anticyclone above the station, which causes local negative inverse barometer effect"*

We clarify that we are not denying that high pressure system lead to a negative sea level.Our study clearly supports the importanceof them and of the inverse barometer effect. The fifth paragraph at the beginning of this answers ("*we clarify that we are not denying that [……]pressure that locally acts according to the inverse barometer effect*".has to be added to the conclusions (page 16, from line 6)  to clarify this.

*:*The reference position is defined as the center of the 5deg wide lat-lon cell where the density of anticyclone centres (blue square in figure new1) has a maximum (same procedure that was adopted for table 2).*

**Answers to the specific comments of reviewer 1.**
The following reviewer's specific comments, are here referred according to page and lines (following the reviewer's list)

**Page 3,line 4. There are surely other important sites on the Mediterranean coastal zone with estimated larger values of SLAs. They should be at least mentioned. Please advise on references provide in General Comment #1.**
The sites where surges have the largest values in the Med Sea are the North Adriatic and the Gulf of Gabes. All other sites where surges have relevant values are represented in this selection of coastal stations. See figures 5 and 6 of Conte, D., and Lionello, P. (2013) Characteristics of large positive and negative surges in the Mediterranean Sea and their attenuation in futureclimate scenarios, Glob. Planet. Change, 111, 159-173, https://doi.org/10.1016/j.gloplacha.2013.09.006, 2013.

**Page 4 line 30 – Page 5 Line 7 and Figs. 2-3: High correlation coefficients between modelled and observed MSLP composites are well-expected, since ERA-Interim re-analysis data are corrected based on the same in situ observations that the authors use for comparisons. In any way, are the input (atmospheric) data further properly validated or are they evaluate in previous studies? Please elaborate. It would be preferable if the authors used comparisons of modelled SLP fields vs. measurements by meteorological stations unassimilated in the modelled ERA data.**
The "OBS" and "MOD" SLP composites are both based on ERA-Interim. The labels "MOD" and "OBS" do not refer to the source of SLP data, but to the criterion used for selecting the members of the samples used to build the composites, that are the time of the SLAs in the hindcast ("MOD") and in the observations ("OBS")(section 3.1, 2$^{nd}$ paragraph). The two samples share only a fraction of members. The similarity of the "MODS" and "OBS" composites, in spite of the small overlap between the corresponding samples, indicates that synoptic conditions leading to surges in the simulations ("MOD") are representative of those leading to the observed surges ("OBS").

**Page 4, line 6-15  and page 6 line 11 and fig.3**
See answer to general comment 5.

**Page 6 line 3-7 and fig.5 This comment seems to be based on a misconception of the inverse barometer effect. In what sense is that an exception? Large negative SLAs are consistent with the very large values of MSLP (huge barometric highs of 1025hPa) in all graphs and over vast areas around all study locations.**
Negative SLAs at Dubrovnik, Thessaloniki, and Tripoli differ from the others because they do not show the presence of a cyclone. The sentence at page 7 from line 1 has been rephrased as "*Figure 4, which considers positive SLAs, shows the presence of a cyclone, which is consequently a permanent feature in the atmospheric circulation leading to large positive SLAs. Also figure 5, which considers negative SLAs, for most stations shows the presence of a cyclone in the basin, except for Dubrovnik, Thessaloniki and Tripoli.*"

**Page 6, line 21-22. This seems like a circumstantial observation and should be backed by wind roses or maps in the area to support the existence of offshore winds over shallow continental shelf.**
See our answer to general comment 4. The columns added to figures 12 and 13 contain maps to support our statement.

**Page 6 Lines 18-19: With an average velocity of translation of the cyclone center close to 32km/hr (Lionello et al., 2016), for a timespan of 44hr (as top in Fig. 5), you have a movement of the low barometric center of about 1536km, which is still very small compared to the distance in Fig. 5 map.**
Actually, we do not understand this comment of the reviewer. At the average speed of 32km/hour a cyclone would cover 768km in one day. This is compatible with the shift of the cyclone centers in the last 24 hours at Iskenderun and Alexandria.

**Page 7 Lines 15-23: This analysis could only be corroborated by correlation to the wind characteristics by PCA method, SOMMS approach and/or other methods of weather pattern identification. It could be omitted if not supported by further analytical comments and results on correlation of SLAs to atmospheric forcing.**
The added columns in figures 12and 13support (in our opinion convincingly) the role of a wind set-up and set-down for stations (mainly Trieste and Gabes) with a long shallow water fetch offshore

**Page 7 Lines 29-31. This sentence needs rephrasing. From "showing that he main…" and on this expression is not an explanation or a conclusive remark but a repetition of the first half sentence.**
The sentence (page 8 from line 29) has been rephrased: "*For example, the tracks of cyclones associated with negative SLAs in Alicante, Toulon and Trieste are similar to those associated with positive SLAs in Thessaloniki, showing that the same cyclone when moving along the main branch of the Mediterranean storm track can produce negativeSLAs at the former stations and positive at the latter*."

**Page 10 Line 31 and Fig. 9: This exactly proves that the inverse barometer effect is mainly responsible for negative SLAs as MSLPs are pretty high over the certain study locations.**
We fully agree that the inverse barometer effect is mainly responsible for negative SLAs. It appears that our manuscript was misunderstood, on this respect. We think that the changes will make clear in the new version what we actually mean. This is emphasized adding to the conclusions (page 16, from line 7) the new paragraph mentioned in our answer to the general comment 5 and explicitly written at the beginning (paragraph 5$^{th}$) of this document.

**Page 11 Line 36 and Fig. 10: 700km are not rendered as large distances in terms of synoptic scale phenomena. Moreover the fact that big distances of the cyclone center do not allow for any influence of the cyclonic low MSLPs to the point-modelled SLAs is proven by Fig. 10, see e.g. green cells in Iskenderun and Alexandria maps. If there was a similar Figure for negative SLAs this would be further strengthened. Or else if such a figure disproves the authors' claims then this kind of analysis should be discarded form the paper.**
Figure 11 (see the text at page 13 from line 8) is not restricted to positive or negative SLA. It shows how the intensity of cyclones is linked to SLAs (both positive and negative) at the station. "*Figure 11 provides a statistical evidence of a teleconnection linking negative SLAs to cyclones whose centers are located thousands of kilometers far away from the station. The green cells in the Iskenderun and Alexandria panels of figure 11show that a cyclone positioned in those areas is linked to negative SLAs at these two stations. Therefore, this figure clearly shows the connection between intensity ofcyclonesin the opposite part of the basin and negative anomalies at Iskenderun and Alexandria*". This paragraph will be added in section 3.5, page 13 from line 32.

**Page 12 Lines 23-25 and Fig. 12.This is probably the case, but further wind data in the surrounding area in the Libyan and Adriatic Seas are needed to be shown in order to prove that.**
The new column in figure 12 clearly shows the presence of winds blowing onshore and offshore and producing positive and negative SLAs , respectively

ALL wording and typoshave been corrected. Thanks for having noticed them.

**Answer to reviewer 2.**

**Reviewer: In this paper, the authors tried to relate the sea level anomalies at 9 stations in the Mediterranean on a climatological basis with cyclonic tracks, cyclone position and intensity and to further analyse this relationship. The paper is well structured. However, I have to admit that I tried very hard to follow all the methodological steps and to understand some explanations. At some points, verification is required. Furthermore, I have many queries concerning the relationship of negative SLA with cyclones.**

We add here below further explanation to better clarify the meaning of our results and the teleconnection linking a negative sea level anomaly to the presence of a cyclone in the opposite part of the basin. Possibly, our text was not sufficiently clear and the meaning of some parts of the manuscript have been misunderstood by the reviewer. Further, we point to some information that is already present in the manuscript, possibly not sufficiently emphasized.

We clarify that we are not denying that high pressure leads to a negative sea level. On this respect, our study confirms the importance of the inverse barometer effect and show that high pressure causes negative SLAs. We have added a new paragraph to the conclusions (page 16 from line 7) to clarify this:
"*We clarify that we are not denying that high pressure leads to a negative sea level. Our study clearly supports the importance of the local action of the inverse barometer effect for both positive and negative SLAs. The link between large negative SLAs and cyclones that is shown in this study does not describe a local effect, but a teleconnection, supported by a statistical analysis and explained by the large scale structure of the SLP fields. The connection between cyclone in the opposite part of the basin and negative SLAS at the station is mediated by the cross basin pressure gradient and the presence of a high that locally acts according to the inverse barometer effect*".

Further, "*The link between large negative SLAs and cyclones is a statistical concept. It is not meant that cyclones are the cause of large negative anomalies, but that the synoptic condition leading to negative anomalies is frequently associated to the present of a cyclone in the opposite part of the basin.*" This clarification has been addedto the conclusions. (page 16, from line 1)

Moreover, the role of anticyclones is confirmed adding a new text after lines 5-8 of the abstract:
"*The inverse barometer effect produces a positive anomaly at the coast near the cyclone pressure minimum and a negative anomaly at the opposite side of theMediterranean Sea, because a cross-basin mean sea level pressure gradient is associated to the presence of a cyclone. This often coincides with the presence of an anticyclone above the station, which causes local negative inverse barometer effect*"

The text that has been added to the manuscript is denoted with slant characters in this reply and is marked with red in the manuscript.

The review contains the request to clarify 10 short points.

**1. Abstract, page 1, line 2: The reviewer writes that the sentence ".. with dynamics involving different factors" is not valid since the authors discuss only the inverse barometer effect.**
**The reviewer thinks that the effect of the wind is also speculated…**

The hindcasts are performed with a dynamical model (HYPSE, Lionello 2005) based on solving the shallow water equations for depth average currents. The model computes the evolution for sea level resulting from the action of surface pressure, wind stress and bottom friction. Therefore, all relevant factors are included in the dynamics leading to the computed SLAs. The inverse barometer effect has the advantage that it can be immediately diagnosed from the SLP field. The residual is due to the wind and eventually to non-stationary, effects. The new versions of figures 12 and 13 contain the composite of the wind fields and show how the intensity of the wind blowing onshore and offshore is associated with the large residuals over shallow water areas. We hope that this clarifies the meaning of the sentence and the role of the wind.

**2. The reviewer (referring to Section 1, page 2, lines 24-29) writes that "the objectives are not clear and robust. In the whole paragraph, the same objective is actually repeated with other words."**

The text quoted by reviewer 2 is repeated here below. The same wording of the text is used, but for sake of clarity in our reply, bullets points are used to split sentences and (in red) the subsections to which the text refersare added. We do not see the reason for the strong criticisms of the reviewer. The text in blue might be consider repeating the first sentence and it could, eventually, be deleted. However, this looks to us to be a question of taste and not a major criticism to the manuscript.

"This study investigates the link of both positive and negative large SLAs along the Mediterranean coastline to the passage of cyclones over the region (figure 1) and describes how SLAs evolve and respond to the presence of cyclones. It includes an analysis:
- of the dynamics of SLAs, (section 3.6)
- of the synoptic patterns associated with them (section 3.2) and
- of variations of these patterns with the position where the SLA occurs(section 3.3)

It aims at contributing arguments for understanding the link between the variability and evolution of the MR storm track and of SLAs. It
- describes position and track of cyclones that are associated with extreme SLA (section 3.3)

and
- shows the link between their intensity and the magnitude of the corresponding SLAs (section 3.5)"

**3. Section 2:the reviewer thinks that, since the hindcastis based on a 2D barotropic model, this allows many simplificationsin the results since the temperature variations are not considered. The reviewer thinks that this is an important limitation and could account for the big differences of SLAs in the observedand simulated time series.**

At page 4, from line 10, in the manuscript we write that
"a) both observations and model results have been preprocessed using a HPF (High-Pass Filter) with a cutoff frequency of 1/30 days (Conte and Lionello (2013) in order to cancel long term components (due to change of mass of the Mediterranean Sea and steric effects) and to isolate the component that is caused by the short term meteorological forcing."
Consequently the comment of the reviewer is, in our opinion, already addressed in the text. The adopted procedure is meant to ensure that our results are not significantly influenced by steric effects.
Further, the discrepancies between observed and simulated SLAs have mainly strong implications for their ranking. At page 5 from line 16 we write that
"In general, the ranking of SLAs in the observed and simulated time series differs substantially. Consequently, the list of the 100 largest observed ("OBS") and of the 100 largest simulated ("MOD") events share only a fraction of events (table 1). Thesmall number of common events is explained by the grouping of the largest SLAs in a relatively narrow range of values, so that small differences in their magnitude may correspond to large differences in their rank. Therefore, inaccuracies of the HYPSE model and of the driving meteorological fields imply substantial differences in ranking between observed and simulated SLAs."

**4. Page 4, line 15: The reviewer asks what mean "depth of thecyclone" means**
A footnote has been added to the text: (page 5)*"The depth of a cyclone is an estimate of the differences between the pressure minimum and the surrounding background value (see Reale and Lionello, 2013 for details on its computation)."*

**5. Section 3.1, page 4, lines 24: The reviewer is surprised about these results and claims that the differences are enormous! He asks us to comment on that and wonders about the reasoning for the hindcast**.

The explanation of the difference is the text above (see our answers to comment 3). The motivation of the hindcast is to provide long time series of sea level at locations where surges are relevant and long observed time series are not available

**6. Sections 3.3-3.4: The reviewer writes that "The association of the SLAs with the density of cyclones is rather arbitrary" and asks a series of questions.**

**why a radius of 20 degs from the coast station is selected for search of MSLP?**
The search radius is a subjective choice, resulting from empirical tests.
Anyway, for positive SLAs, the outcomes of the search depends very weakly on this parameter. In fact, the resulting cyclone centers are closely grouped around the station at a distance much smaller than 20degrees (see figure 6). For negative SLAs a small search radius would miss to detect the presence of a cyclone in the basin and would not allow the analysis of the teleconnection between cyclones and negative SLAs.

**Why the computation of the relative frequency is based on 10 deg radius?**
The reference point for searching a cyclone center and computing the frequencies in table 2 and 3 is not the station, but the point around which the cyclone centers concentrate. Therefore, the former search radius and this parameter follow different logics. Ten degrees are about 850 km in the zonal direction in the central areas of the Mediterranean Sea. The average size of cyclones in the Mediterranean is about 500km, of course with non-negligible geographical differences (see table 7 of Lionello et al. 2006 after Trigo et al, 1999) and they move at the average speed of about 180km in six hours (time step of the ERA-Interim data). Consequently, the 10degrees radius is meant to detect cyclones passing close to the reference point at the time of the SLAs.

Trigo, I. F., Davies, T. D., &Bigg, G. R. (1999). Objective climatology of cyclones in the Mediterranean region. J. Climate, 12, 1685–1696.
Lionello P., Bhend J., Buzzi A., Della-Marta P.M., Krichak S., Jansà A., Maheras P., Sanna A., Trigo I.F., Trigo R. (2006). Cyclones in the Mediterranean region: climatology and effects on the environment. In P.Lionello, P.Malanotte-Rizzoli, R.Boscolo (eds) Mediterranean ClimateVariability.Amsterdam: Elsevier (NETHERLANDS), 325-372

**Why a time step of 10 days is selected?**
This step is used for extracting independent SLP fields from the hindcast in order to estimate the probability that a cyclone is present. In general, the requirement that samples are independent is essential for a correct estimate to avoid double counting the same cyclone several times. Lionello et al 2016 show that in the Mediterranean cyclones lasting more than 5 days are extremely rare. Therefore, this step ensures that the climatological probability is estimated using independent samples and every cyclone is counted only once.

**Why the reference point is located in the Ionian sea based on a subjective criterion?**
First, it is important to note that the reference point has been located subjectively only in Iskenderun for negative SLAs. In all other cases "The reference position is the center of the 5deg wide lat-lon cell where the density of cyclone centers has a maximum." (lines 8-10 at page 9). For negative SLAs at Iskenderun (In this case cyclone centers are rather sparsely distributed) this criterion would locate the reference center at the eastern boundary of the map, downstream of the Mediterranean region. To avoid this, the secondary maximum (largest value after the actual maximum) has been used (which is the point located in the Ionian basin). The new text explaining this the text "*For negative SLAs at Iskenderun, where this criterion would locate the reference point at the eastern boundary of the map, the second largest maximum value (in the middle of the Ionian Sea) has been used.*" has been added at page 9, lines 10-11

**7. The reviewers asks "why the negative SLAs are related with cyclones and not with anticyclones. This seems a more realistic thoughtand approach**. "

A new figure and a table have been produced to describe the role of anticyclones
*"Figure 8 shows the centers of anticyclones at the time of negative SLAs. It is made following the same procedure that has been used for figure 7, which refers to cyclones. It reveals the location of centers of anticyclones in the areas where figure 5 shows high pressure systems. Anticyclones are actually concentrated around the stations, with the exception of Gabes and, to a lesser degree, Trieste, where the wind effect is much larger than the inverse barometer effect and anticyclones play a minor role. Therefore, negative SLAs are linked to the presence of a high pressure around the station. This is necessarily true for most stations, because of the inverse barometer effect. However (see table 4), the probability to find an anticyclone at a distance*

*lower than 10 degrees from the reference position\* at the time of negative SLAs is significantly larger than the climatological value only for three stations (Toulon, Thessaloniki, Iskenderun). On the contrary, in Gabes, the absence of an anticyclone is linked to negative SLAs (and this is justified by the dominant role of the wind at this station). The link with the presence of a cyclone in the part of the basin opposite to the station (table 3) is stronger than what shown in table new1 for anticyclones."* This explanation has been added to the manuscript at the end of section 3.4.

Further, an objective of this study is to show that a robust teleconnection, which is supported by a statistical analysis, links negative SLAs at some stations to the presence of a cyclone in the opposite part of the basin. This link does not describe a local effect. In fact, the connection between cyclone in the opposite part of the basin and negative SLAS at the station is mediated by the cross-basin pressure gradient and the presence of a high that locally acts according to the inverse barometer effect. We anticipated at the beginning of this reply that the new text that clarifies has been added to the manuscript (page 16, from line 1).

\*:*The reference position is defined as the center of the 5deg wide lat-lon cell where the density of anticyclone centres (blue square in figure new1) has a maximum (same procedure that was adopted for table 2).*

**8. The reviewer is "not convinced about the reliability of the results in sections 3.3 and 3.4". According to him/her "Many findings are speculated and not verified. The positive SLAs could be related with frontal systems that are not considered in this study.**

The arguments why the reviewer is not convinced are not given. It is therefore a bit difficult to propose changes or to argue against her/his reluctance to accept the content of these sections. She/He does not say which "findings are speculated and not verified". The results in table 2 and 3 are checked for statistical significance at the 95% confidence level. Mid latitude cyclones are characterized by the present of cold, warm, and occluded fronts, therefore, when considering a cyclone, the action of the fronts associated with it are included.

**9. Section 3.5, page11, line 4: The reviewer asks "why a linear regression is used? A lag correlation should be attempted sincethe effect of cyclones on the storm surges is not always instant."**

A linear regression is the simplest tool to model the relationship between a dependent variable (in this case the SLAs) and an explanatory variable (in this case the SLP minimum). The two variables are sometime called predictand and predictor respectively. In this case, a statistically significant linear relation is found and it shows that the SLA levels are linked to the intensity (minimum SLP) of the cyclones. We agree that this approach ignore the possibility of delayed effects, and therefore may conceptually underestimate the strength of the link (that could be stronger, not weaker, than what we have found). Therefore, this approach based on linear regression is successful and we have no reason for rejecting it. Certainly, other approaches to reach the same goal are possible.

**10. Section 3.6: The term "dynamics" is not relevant since there is no discussion on the flow regime.**
Dynamics is a branch of physical science and subdivision of mechanics that is concerned with the motion of material objects in relation to the physical factors that affect them (Encyclopedia Britannica). In section 3.6 we describe the resulting position of the sea surface (SLA) in relation to the sea level pressure and the wind fields that cause its motion. In our view the term dynamics is justified.

**Answer to reviewer 3.**

**Reviewer 3: The manuscript presents an interesting analysis of the relationship between large sea level anomalies (SLA) and cyclones in the Mediterranean Sea. The assessment is based on the 100 largest positive and negative SLA from a 22y hindcast at 9 coastal sites along the Mediterranean coast; and the characteristics of cyclones that occurred in the Mediterranean region during the hindcast period. The links between the largest SLA and the cyclones intensities and the location of their centers are statistically analyzed in order to associate the SLA to the synoptic atmospheric conditions that generate them. For most of the stations, the results show that large positive SLA are caused by the presence of a cyclone in the area of the station i.e. the large positive SLA are caused by the inverse barometer effect. On the other hand, some large negative SLA arise from the presence of a cyclone at the other side of the Mediterranean basin, which generates a MSLP gradient along the basin. The manuscript is well written and structured; the topic is relevant and the results and findings are interesting and relevant. However, I think that there are few important points that should be addressed and I also have few minor comments.**

We thank Reviewer 3 for her/his comments on our manuscript. Here below are our replies to her/his major concerns (see items 1 to 5 below, in bold) and the description of the added material that we have produced for properly addressing them. The text that has been added to the manuscript and is denoted with slant characters in this reply and is marked with red in the manuscript. Some of these concerns/suggestions are shared by reviewer 1, particularly on the need of more explanations describing a) the role of the wind and b) the link to anticyclones. In fact, some new material to address these two issues that is described in this reply is also present in our answers to reviewer 1:

1) **Reviewer: My major concern about the study is related to the variability of the events sample. First, it is not clear to me why 100 events were selected; is this a subjective decision? From the results shown in Table 2 and 3 it is clear that the different synoptic MSLP fields (i.e. both the cyclones associated to different regions and those not assigned) caused both large positive and negative SLA in most of the stations. However, all events are analysed as a single sample and in some cases the SLA or MSLP fields associated to all events are averaged. This can "hide" or weaken the links between the SLA and the MSLP fields due to the variability of the sample. In my opinion, the results would benefit from clustering the events sample in order to reduce the variability, e.g. by a principal component analysis of the MSLP fields associated to the SLA, and performing a separate analysis to each cluster. At least, this issue should be addressed in the discussion.**
We admit that *"The selection of 100 hundred events is a subjective decision. Considering that the hindcast covers 22 years, this corresponds to an approximate average of 5 events per year. In the case of Venice this is close to the 80$^{th}$ percentile of the surge events (Lionello et al, 2012\*). Empirical tests have shown that results do not appreciably change using a smaller sample."* This sentence has been added to the data and methods section (page 4, from line17).
Splitting the samples in subsets using statistical techniques such as PCA or clustering is certainly a possibility. However, in our study *"the internal variability of the sample is explored considering the analysis of the cyclogenesis, which allow to distinguishthe different evolutions of cyclones. We have adopted this process-oriented approach. In our opinion, it is very plausible that PCA or clustering of the trajectories would have produced very similar outcomes.We admit that this approach might hide some aspects of the internal variability of the sample related to different synoptic patterns at the time of the SLAs, which might be worth to explore in a future studies for those stations where this issue would eventually result significant."* This paragraph has been added to the "Discussion and conclusion" section, page 16 from line 23

\*Lionello P, Cavaleri L, Nissen KM, Pino C, Raicich F, Ulbrich U (2012) Severe marine storms in the Northern Adriatic: Characteristics and trends. *PhysChem Earth,* 40-41:93-105, doi:10.1016/j.pce.2010.10.002

2) **Reviewer: Following the previous comment, it is not clear in some cases how the presented composites are generated. For example, are composites of Figures 4 & 5 showing average values of MSLP fields from all events? Composites shown in Figures 11 and 12 show the total anomaly, is this**

**the sum of all SLAs? Giving the range of the magnitude of the selected SLAs could also give an idea of the variability within the events.**

We confirm that the composites of Figures 4 & 5 show (for each station) average values of MSLP fields from all events. Analogously, composites in Figures 12 and 13 show the average value of the anomaly and its components (Inverse barometer effect and residual). A further column has been addedto figures 12 and 13 with the composites of the wind fields. The text (page 6, from line 21) reports that "The panels of figures 4 and 5, based on the 100 largest positive and negative SLA, respectively, show the ERA-Interim MSLP composites 48 hours (left column), 24 hours (mid column) before and at the time (right column) of the SLA maxima."

**3) Reviewer:I would suggest to add more details about the cyclone characteristics because it is unclear what is the largest MSLP that can be associated to a cyclone center, or what the differences between shallow and depth cyclones are..**

Indeed, the adopted tracking algorithm provides further information that may be useful to add. We have prepared four tables (Tables SuM1-SuM4, see below) that show for each cyclogenesis area the mean values of the central SLP minimum and of the depth of the cyclone with the respective standard deviations. If the reviewer thinks that they are useful, we propose to add them in the supplementary material. The results show that cyclones generated over the Atlantic are deeper and with a lower central SLP minimum than those generated in other areas, consistently with the known climatology (Lionello et al.2016)*

*Lionello, P., Trigo, I.F., Gil, V.,Liberato, M.L.R., Nissen, K., Pinto, J.G., Raible, C.,Reale,M., Tanzarella, A., Trigo, R.M., Ulbrich, S. and Ulbrich, U.:Objective Climatology of Cyclones in the Mediterranean Region: a consensus view among methods with different system identification and tracking criteria, Tellus A, 68, 29391, https://dx.doi.org/10.3402/tellusa.v68.29391, 2016.

4) **Reviewer: Regarding negative SLAs, the analysis is only focused on their correlation with the presence of a cyclone in the opposite region of the Mediterranean basin, but I would imagine that large negative SLA will be highly correlated to high pressure systems (which can be supported by the large number of events not assigned to cyclones reported in Table 3).**

The new figure 8, which describes the role of anticyclones, and a new table have been produced."*Figure 8 shows the centers of anticyclones at the time of negative SLAs. It is made following the same procedure that has been used for figure 7, which refers to cyclones. It reveals the location of centers of anticyclones in the areas where figure 5 shows high pressure systems. Anticyclones are actually concentrated around the stations, with the exception of Gabes and, to a lesser degree, Trieste, where the wind effect is much larger than the inverse barometer effect and anticyclones play a minor role. Therefore, negative SLAs are linked to the presence of a high pressure around the station. This is necessarily true for most stations, because of the inverse barometer effect. However (see table 4), the probability to find an anticyclone at a distance lower than 10 degrees from the reference position\* at the time of negative SLAs is significantly larger than the climatological value only for three stations (Toulon, Thessaloniki, Iskenderun). On the contrary, in Gabes, the absence of an anticyclone is linked to negative SLAs (and this is justified by the dominant role of the wind at this station). The link with the presence of a cyclone in the part of the basin opposite to the station (table 3) is stronger than what shown in table new1 for anticyclones.*" This explanation has been added to the manuscript at the end of section 3.4.*

Further, we will extend the sentence at lines 5-8 of the abstract as it follows: *"*The inverse barometer effect produces a positive anomaly at the coast near the cyclone pressure minimum and a negative anomaly at the opposite side of theMediterranean Sea, because a cross-basin mean sea level pressure gradient is associated to the presence of a cyclone. *This often coincides with the presence of an anticyclone above the station, which causes local negative inverse barometer effect"*

We clarify that we are not denying that high pressure leads to a negative sea level because of the inverse barometer effect and our study clearly supports its importance. The following paragraph has been added to the conclusions (page 16, from line 7) to clarify this.

"*we clarify that we are not denying that high pressure leads to a negative sea level. Our study clearly supports the importance of the local action of the inverse barometer effect for both positive and negative SLAs. The link between large negative SLAs and cyclones that is shown in this study does not describe a local effect, but a*

*teleconnection, supported by a statistical analysis and explained by the large scale structure of the SLP fields. The connection between cyclone in the opposite part of the basin and negative SLAS at the station is mediated by the cross basin pressure gradient and the presence of a high pressure that locally acts according to the inverse barometer effect"*. This paragraph has been added to the "Discussion and Conclusion section".

*: *The reference position is defined as the center of the 5deg wide lat-lon cell where the density of anticyclone centers (blue square in figure new1) has a maximum (same procedure that was adopted for table 2).*

5) **Reviewer: From the analyses presented it is very difficult (or even impossible) to observe the effects of the wind set-up discussed by the authors. For instance, the wind effects can be represented by adding MSLP gradients maps.**
Following this suggestion, two columns with the wind composites at the time when SLAs are largest have been added to figures 12 and 13, with the following text commenting them: "*The action of the wind is evident in the fourth column of figures 11and 12, which show the composites of the wind fields at the time when the SLAs are largest anomaly. In these maps, the presence of a strong wind blowing towards the coast (fig.11, positive SLAs) or offshore (fig.12, negative SLAs) is consistent with the large residuals at Trieste, Tripoli and Gabes. For positive SLAs the wind is also present in correspondence with the residuals (which are smaller than in the previous stations) at Alexandria, Iskenderun and Thessaloniki.*" These sentences have been added at the end of section 3.6.

Thanks for correcting typos. We have only one comment on the use of "an" (which has been maintained) before the abbreviation "SLA". This should depend on the pronunciation of the abbreviation and not on its actual first letter. We would leave the choice to the technical editing of the paper. There are however, two minor comments that require to be addressed.

**P3-L35. Is there any reason for selecting a time window of 120h? e.g. is this value based on any previous study on the duration of storm surges in the region?**

A footnote (page 4) has been added: "*This period has been selected to ensure independence of the events. considering the whole Mediterranean region, {Lionello et al. (2016) show thet cyclones lasting more than 5 days are extremely rare. Considering the specific situation of the Adriatic Sea, it is meant to avoid the superposition with seiches triggered by previous events, which have a period of about 22 hours and an attenuation of about 10% at each cycle Lionello et al. (2006b).*"

**Reviewer: Section 3.1. In this section, I am missing the bias between SLAs from the modelled and observed datasets in order to give an idea of the model performance (also e.g.RMSE).**
The model validation has already been discussed in Conte and Lionello 2013. The following text has been added to section 2 (page 3, from line 24) data and methods to summarize these former results: "*The simulation describes well the large SLA values in Northern Adriatic sea and describe the difference between this sub-regionalpeak and the rest of the coastline.Unfortunately,lack of data prevent model validation along the African coast. In general the model underestimates large SLAs, with a tendency to perform worse in the western Mediterranean than along the rest of the coast and to perform percentwise better for negative than positive SLAs. Tide gauge data for validation are available only in four of the stations considered in this study (Alicante, Toulon, Trieste, Dubrovnik). Percent rms error on large SLAs is less than 10% for Toulon and Trieste, and in the range 30-40% for Alicante and Dubrovnik*"

| Station (SLA+) | MSLP$_{ATL}$ | MSLP$_{AFR}$ | MSLP$_{WM}$ | MSLP$_{EM}$ | MSLP$_{AsEU}$ |
|---|---|---|---|---|---|
| ALICANTE | 998±6 | 1005±6 | 1005±5 | | |
| TOULON | 997±5 | 1003±5 | 1003±6 | | |
| TRIESTE | 992±8 | 999±6 | 1002±5 | | |
| DUBROVNIK | 996±7 | 997±4 | 1001±5 | 1004 | 1010 |
| THESSALONIKI | 999±3 | 1001±3 | 1002±5 | 1004±6 | 999±8 |
| ISKENDERUN | 999±3 | 1004±3 | 1001±4 | 1005±4 | 1002±9 |
| ALEXANDRIA | 999±2 | 1004±4 | 1004±4 | 1006±4 | 1006±5 |
| GABES | 1000±5 | 1006±4 | 1007±7 | | |
| TRIPOLI | 1002±3 | 1005±5 | 1005±6 | 1006±4 | 1012 |

Table SuM1: This table considers positive sea level anomalies and show the mean values (with standard deviation) of the central pressure minimum considering for each station (rows) the different cyclogenesis areas (columns). Values are in hPa. Blank cells denote absence of cyclones originated from the corresponding area. Obviously, the standard deviation is not provided when only one cyclone is present. The areas (Atl, Afr, WM, EM, AsEu) are shown in figure 1

| Station (SLA+) | Depth$_{ATL}$ | Depth$_{AFR}$ | Depth$_{WM}$ | Depth$_{EM}$ | DepthAsEU |
|---|---|---|---|---|---|
| ALICANTE | 2090±509 | 1447±477 | 1534±523 | | |
| TOULON | 2284±541 | 1472±756 | 1750±475 | | |
| TRIESTE | 2563±756 | 1858±494 | 1819±542 | | |
| DUBROVNIK | 2385±639 | 1882±318 | 1911±517 | 1578 | 2504 |
| THESSALONIKI | 2178±355 | 1714±283 | 1929±450 | 1709±473 | 2192±914 |
| ISKENDERUN | 2189±500 | 1438±420 | 1914±324 | 1570±329 | 1593±600 |
| ALEXANDRIA | 2162±290 | 1390±481 | 1650±350 | 1463±407 | 1804±1000 |
| GABES | 1894±497 | 1191±358 | 1478±511 | | |
| TRIPOLI | 1937±255 | 1471±392 | 1690±452 | 1476±322 | 856 |

Table SuM2: Same as table SuM1, except it refers to the depth of the cyclones. Values are in Pa

| Station (SLA-) | MSLP$_{ATL}$ | MSLP$_{AFR}$ | MSLP$_{WM}$ | MSLP$_{EM}$ | MSLPAsEU |
|---|---|---|---|---|---|
| ALICANTE | 1000±4 | 1002±3 | 1003±5 | 1005±7 | 997±7 |
| TOULON | 1000±3 | 1003±3 | 1006±5 | 1009±1 | 1012 |
| TRIESTE | 1000±5 | 1002±5 | 1003±6 | 1007±6 | 1005±8 |
| DUBROVNIK | 999±7 | 1004±4 | 1004±4 | | |
| THESSALONIKI | 1001±6 | 1005±5 | 1003±6 | | |
| ISKENDERUN | 1001±6 | 1004±5 | 1004±5 | 1008 | 1006 |
| ALEXANDRIA | 1000±4 | 1004±5 | 1001±5 | 1008 | |
| GABES | 1000±4 | 1003±5 | 1003±5 | 1003 | 986 |
| TRIPOLI | 1000±8 | 1006±4 | 1008±7 | | |

Table SuM3: Same as table SuM1, except it refers to negative sea level anomalies (Values in hPa)

| Station (SLA-) | Depth$_{ATL}$ | Depth$_{AFR}$ | Depth$_{WM}$ | Depth$_{EM}$ | DepthAsEU |
|---|---|---|---|---|---|
| ALICANTE | 1998±425 | 1596±242 | 1709±475 | 1544±420 | 2282±794 |
| TOULON | 1894±357 | 1353±364 | 1706±441 | 1026±335 | 856 |
| TRIESTE | 1973±597 | 1628±511 | 1815±547 | 1473±455 | 1424±361 |
| DUBROVNIK | 2034±688 | 1368±404 | 1486±376 | | |
| THESSALONIKI | 2039±604 | 1300±502 | 1761±490 | | |
| ISKENDERUN | 1969±358 | 1387±541 | 1660±403 | 1065 | 1386±201 |
| ALEXANDRIA | 1942±459 | 1457±419 | 1794±577 | 1065 | |
| GABES | 2045±412 | 1597±419 | 1851±337 | 1541 | 2994 |
| TRIPOLI | 2088±473 | 1293±256 | 1280±574 | | |

Table SuM4: Same as table SuM1, except it refers to negative sea level anomalies and to the depth of the cyclones (Values in Pa)

---

## Referee Report (RR1)

The authors of the #nhess-2019-6 submitted paper have made a great effort to address my main concerns (which basically coincided with the comments of the rest two Reviewers). Authors have now clarified any misrepresentation of their notion of the negative SLAs drivers (i.e. tele-connected barometric lows or the immediate inverse barometer effect over the area of focus) leaving little space for any misconception in the revised text. The impact of the cross-basin SLP gradients due to the presence of cyclones is now better explained and the newly added wind patterns strongly back up the previously uncorroborated authors' claims. The paper is now better structured and clearly re-written and should be **accepted for publication**. Only minor issues are left to deal with, subjected to a last **minor revision**, mostly related to improvement of the background literature review and a few easy-to-do clarifications. In the following, I present a few last comments together with some typos corrections.

**General Comments:**

**1) Introduction; Page 2:**

The authors insist not to refer to proposed literature, probably to save space and unnecessary info in their paper. It is reasonable that out of the indicative bunch of proposed papers, Bengtsson et al. (2006) concerns only storm tracks under climate change, Calafat et al. (2012) deals with hindcast modelling of only Mean Sea Level variations, Campins et al. (2006) only deal with storm tracks in climate change, Vousdoukas et al. (2017) and Fernández-Montblanc et al. (2019) deal mainly with other divergent issues, and thus all could be indeed left out from the authors' literature review.

However that is not the case for the following: a) Marcos et al. (2011) refer to storm surge modelling under both a control run for 1950–2000 and hindcasts from 1958 to 2001 forced by dynamically downscaled ERA40 reanalysis data linking between atmospheric features and surges; b) Makris et al. (2016) present a lot of information about the prevailing synoptic systems over eastern Mediterranean, including hindcast analysis of a 50-yr reference period up until 2000, together with several comments on the relation of extreme storm surges to atmospheric conditions (wind patterns and synoptic conditions) in the east-central Mediterranean; c) Vousdoukas et al. (2016) also present results of modelled surges on the Mediterranean coastline for a baseline period from 1970 to 2000 validated against simulations driven by ERA-Interim atmospheric forcing, indicating good skills.

At least these three articles should be mentioned in the state-of-the-art of the Introduction as they are important cases dealing with similar issues presented in the submitted paper.

Furthermore, two more crucial papers of Androulidakis et al. (2015) and Ullmann et al. (2007) are only listed in the References but not cited/discussed in the main text of the paper. They should be mentioned in the text with a concise comment of their findings, as they are important background literature on the issues covered by the authors.

Marcos, M., Jordà, G., Gomis, D., Pérez, B. (2011). Changes in storm surges in southern Europe from a regional model under climate change scenarios. *Glob. Planet. Change*, 77(3): 116–128.

Makris, C., Galiatsatou, P., Tolika, K., Anagnostopoulou, C., Kombiadou, K., Prinos, P., Velikou, K., Kapelonis, Z., Tragou, E., Androulidakis, Y., Athanassoulis, G., Vagenas, C., Tegoulias, I., Baltikas, V., Krestenitis, Y., Gerostathis, T., Belibassakis, K. and Rusu, E. (2016). Climate Change Effects on the Marine Characteristics of the Aegean and the Ionian Seas. *Ocean Dynamics*, 66(12): 1603–1635.

Vousdoukas, M.I., Voukouvalas, E., Annunziato, A., Giardino, A., Feyen, L. (2016). Projections of extreme storm surge levels along Europe. *Clim. Dyn.*, 47: 3171–3190.

**2) old Comment 3; Pages 7-8:**

In the revised paper, the authors have made a decent effort to transform their previously rather untenable claims about negative SLAs' association with cyclonic motions in the atmosphere on sites practically far away at the opposite side of the basin. This clearly presented in the new "Discussion and Conclusions" section, thus any misinterpretation is avoided in the new text version. Yet, a minor further clarifying comment referring also to findings of Figs. 3-5 should also be added i.m.o., similar to the following: "*In general, the presence of a cyclone or a barometric low system is not considered as the immediate cause of negative SLA events, but a probable secondary driver of them. Negative SLAs can be only collaterally associated with cross-basin SLP gradients, with more certainty especially in cases where winds are driven by near cyclone centers, since correlation of parameters does not necessarily prove causation between the examined features.*"

**3) old Comment 5**

My point was that there exist cyclogenetic centers (or areas with barometric lows in general) outside of the Mediterranean window shown in the paper that may influence the study area in terms of SLA response to them. Therefore it should be at least mentioned, and further clarified that "*teleconnection" is a notion referring to global scales (or at least usually larger regions than the one examined in the paper) that needs bigger windows of application, in order to pertain crucial dynamic cyclogenesis centers surrounding the Mediterranean or in the vicinity of it*. A good example of a typical teleconnection index is the NAO, whose effect on weather patterns and oceanographic features of e.g. the Mediterranean needs analysis covering large parts of the Atlantic Ocean and even North Sea regions.

**Specific Comments:**

**Figure 11 and Page 13:**

The authors refer to linear regression coefficient as a statistical measure to relate teleconnection phenomena to negative SLAs. Is it the classic simple linear regression $r$ coefficient? If so, shouldn't it better be a statistical coefficient of determination, e.g. $r^2$ = the Pearson's product-moment coefficient,

in order to show a more robust approach on the dependence or association in the statistical relationship of moving cyclone centers to far-away negative SLAs?

**Page 3, Lines 18-25:**

The authors refer to a hindcast validation of HYPSE model in their 2013 paper, but if the paper presents new runs with the newest parallel HYPSE version does that mean that new simulations results are not validated? That is because the parallelization of codes is known to induce some numerical instability in results, besides their beneficial reduction of computational times.

**Page 4, Lines 6-7:**

In the last sentence, please correct the expression. The authors do not have in their possession long-term tide-gauge data for sea level elevation. They are available by National Hydrographic Services.

**Page 13, Lines 1-3:**

Referring to the contrast of Fig. 7's dense cyclone centers in Fig. 10 e.g. *a* or *b* graphs is the respective MSLP difference of marginally 5-10hPa large enough to support the authors' claims?

**Page 14, Lines 8-22:**

Please configure text about central/right column are referred, according to the changed Figure 12-13 new four columns.

**Technical/Editorial Comments:**

Page4, Line30: correct to "short lived features"

Page4, Line32: Put full-stop mark after "Europe"

Page13, Line32: correct to "the station show"

Page15, Line12: delete "anomaly" after "largest"

Page16, Line17: delete one "area"

Page16, Line20: correct to "Wm is the main overall…"

Page16, Line29: correct "causing"

---

## Author Response (AR2)

Dear Editor:

Thanks for your handling of this manuscript. We attach a revised version, where the minor comments of the reviewer are addressed (additional text is marked red and deleted text is crossed).

Precisely, considering the reviewer's general comments:

1) Introduction; page 2…. The reviewer insists adding some bibliographic references and we have accepted her/his suggestions. We have added the suggested bibliographic items and integrated with some others on the same subject. Androulidakis et al. (2015) and Ullmann et al. (2007) are now referred in the text (see page 2, line 18-19 and 25-27)

2) Old comment 3; pages 7-8. The explanation suggested by the reviewer is already present in the 3rd and 4th paragraphs of "Discussion and Conclusion". We have further reinforced it adding the text at page 15, line 30)

3) Old comment 5. We have added a footnote when teleconnection are first mentioned in the paper at page 13 where we write that "Here teleconnection does not refer to a mechanism acting at global scale, but across different areas of the Mediterranean region" and added the connotation "synoptic scale teleconnection" at page 16 line 5

Considering the reviewer's specific comments

✓ Figure 11 and Page 13: here statistical significance is considered when the null hypothesis of no trend can be discarded at the 90% confidence level. Other slightly different approaches are obviously possible.

✓ Page 3, Lines 18-25: we admit the text was confusing. No new simulation was performed for this study. We used the same simulations as in the 2013 paper. The text has been corrected (see page 3, lines 21 and 24-25)

✓ Page 4, Lines 6-7: the sentence has been modified (see page 4, lines 6-7)

✓ Page 13, Lines 1-3: The residual in the third column of figure 12 shows to which extent other factors beside the inverse barometer effect are at play. A short comment has been added (see 13,line 30)

✓ Page 14, Lines 8-22: The text has been corrected to describe the four columns of figures 12 and 13 (see page 14, lines 4-7).

All editorial comments have been corrected.

Best regards
Piero Lionello on the behalf of all authors